# Explainable AI-driven diagnosis model for early glaucoma detection using grey-wolf optimized extreme learning machine approach

Debendra Muduli[1], Santosh Kumar Sharma[1], Sujata Dash[2]*, Bernardo Lemos[3,4], Saurav Mallik[3,4]*

1 Department of Computer Science and Engineering, C.V. Raman Global University, Bhubaneswar, Odisha, India, 2 Department of Information Technology, School of Engineering and Technology, Nagaland University, Meriema, Nagaland, India, 3 Department of Environmental Health, Harvard T H Chan School of Public Health, Boston, Massachusetts, United States of America, 4 Department of Pharmacology & Toxicology, University of Arizona, Tucson, Arizona, United States of America

* smallik@arizona.edu, sauravmtech2@gmail.com (SM); sujata238dash@gmail.com (SD)

## Abstract

Glaucoma is a leading global cause of blindness, making early detection essential. This paper introduces GlaucoXAI (Glaucoma Explainable AI), an advanced computer-aided diagnosis (CAD) model that integrates machine learning and explainable AI for glaucoma detection using retinal fundus images. The proposed model consists of four stages, including preprocessing, feature extraction, dimensionality reduction, and classification. Initially, features are extracted using the fast discrete curvelet transform with wrapping (FDCT-WRP) to obtain curve-type features. During the next stage, principal component analysis (PCA) and linear discriminant analysis (LDA) are combined to reduce the dimensionality of the feature matrix, followed by a classification stage employing an improved grey wolf optimization (IMGWO) with an extreme learning machine (ELM) to optimize the weight and bias to reduce the overfitting of the model. The model has been experimented with two publicly available datasets named G1020 and ORIGA. The model has achieved 93.87% accuracy on G1020 and 95.38% on ORIGA, outperforming existing methods. The 10×5-fold stratified cross-validation (SCV) with explainable AI enhances the interpretability of models and improves clinician trust. Overall, the proposed approach offers accurate, efficient, and explainable glaucoma diagnosis, potentially supporting ophthalmologists in early disease detection.

## Author summary

Glaucoma is one of the leading causes of irreversible blindness worldwide, yet it often remains undetected until significant vision loss has occurred. In this study, we developed GlaucoXAI, an explainable artificial intelligence (AI) model

which permits unrestricted use, distribution, and reproduction in any medium, provided the original author and source are credited.

**Data availability statement:** The datasets and code used in this study are available at our GitHub repository: https://github.com/Santosh-84ss/IMGWO.

**Funding:** The author(s) received no specific funding for this work.

**Competing interests:** The authors have declared that no competing interests exist.

that helps detect glaucoma early from retinal fundus images. Unlike conventional black-box AI systems, GlaucoXAI combines advanced image analysis with explainable AI methods to make its predictions more transparent and trustworthy to clinicians. The model uses a hybrid approach integrating feature extraction, dimensionality reduction, and optimized neural learning to enhance accuracy and speed. Tested on two publicly available datasets (G1020 and ORIGA), GlaucoXAI achieved over 93–95% accuracy, outperforming existing models. Importantly, it generates visual explanations that highlight regions of the eye most responsible for its predictions, supporting ophthalmologists in verifying AI-driven results. This work demonstrates that AI can be both accurate and interpretable, marking a step toward reliable clinical tools for early glaucoma diagnosis and better patient care.

## 1. Introduction

Recently, ethical concerns regarding transparency in AI have emerged in the healthcare sector. The lack of trust in AI models arises from their opaque functioning, highlighting the need for simplification [1]. Explainable AI (XAI) techniques are methods designed to clarify how AI models work and how they make predictions [2]. Health information collected from individual users can aid ophthalmologists in identifying diseases more effectively [3]. These models allow for continuous health monitoring through healthcare tools such as smartphones and wearable devices, empowering individuals to manage their health proactively.

Computer-aided diagnosis (CAD) for ocular diseases is widely used because it offers significant advantages, including rapid and precise large-scale screening, which helps alleviate the workload of physicians during routine clinical tasks [4]. Glaucoma is a chronic eye condition characterized by damage to the optic nerve (ON) and visual impairment, often evident through the cupping of the optic to disc ratio (OCR) and degeneration of ON fibers [5,6]. Reducing intraocular pressure (IOP) is a well-established and evidence-supported method for treating open-angle glaucoma (OAG) [7,8]. Timely diagnosis and effective management of IOP are crucial for maintaining a high quality of life, especially in aging populations.

Changes in eye structure due to glaucoma often occur before any noticeable functional effects. Therefore, it is essential to classify these structural changes for early detection of the disease. The primary method for glaucoma detection involves examining color fundus images, which can reveal signs of ON damage associated with glaucoma, such as an increased cup-to-disc ratio (CDR), rim thinning, notching, cupping, disc hemorrhage, and irregularities in the retinal nerve fiber layer (RNFL). Additionally, optical coherence tomography (OCT) is considered an effective method for characterizing glaucoma both qualitatively and quantitatively [9]. OCT primarily focuses on the optic nerve head and macula, providing high sensitivity and specificity in detecting preperimetric glaucoma [10]. Various OCT parameters, including disc topography and layer thickness measurements, are effective in capturing

glaucomatous structural changes [11,12]. It is advisable to use OCT information from the overall optic disc and macula as part of a comprehensive diagnostic strategy for identifying different types of glaucoma.

Significant advancements have been made in machine learning technologies, particularly in deep learning, leading to the development of new models for the automated diagnosis of eye diseases like glaucoma. However, it is important to note that existing machine learning (ML) models in these studies have typically focused on just one category of images to differentiate between glaucoma and healthy eyes. This approach differs significantly from the thorough clinical examinations performed by ophthalmologists, resulting in a limited number of ML techniques that utilize fundus images specifically for glaucoma detection.

Machine learning methods have frequently been used to identify various eye conditions, including glaucoma [13], diabetic retinopathy (DR) [14], age-related macular degeneration (AMD) [15], and various retinal issues [16]. Additionally, Retinal Fundus Images (RFIs) serve as a valuable tool for identifying numerous non-ocular conditions, including Type II diabetes [17,18], anemia [19], and cardiovascular risks [20]. In the context of glaucoma classification, various imaging modalities and clinical examinations are employed, such as RFIs [21], OCT [22], and visual field tests (VFTs) [23]. Despite the availability of multiple approaches, fundus imaging remains the most prevalent and cost-effective technique for widespread screening of various retinal diseases [24].

This study aims to enhance result optimization by combining multiple feature selection techniques with the Extreme Learning Machine (ELM) algorithm [25]. The approach employs an Integrated Grey Wolf Optimizer (IMGWO) to achieve improved classification accuracy. Furthermore, the study enhances feature normalization and reduction by using a hybrid method that combines Principal Component Analysis (PCA) and Linear Discriminant Analysis (LDA). Additionally, the study utilizes an IMGWO-ELM algorithm for classifying glaucoma, thereby improving the accuracy of the results.

Recently, the implementation of deep learning methods in clinical settings has been limited by certain constraints. A primary concern is that deep learning approaches focus solely on the input images and their resulting outputs, offering little insight into how information moves through the internal layers of the network. In critical situations, such as fundus imaging, understanding the rationale behind the network's predictions is essential to ensure accurate estimations. Consequently, there is a growing interest in Explainable AI (XAI) to explore opaque deep learning models within the medical domain. XAI techniques enable researchers, developers, and end-users to comprehend deep learning models and articulate their decisions in a way that is understandable to humans. Medical consumers increasingly demand explainability in deep learning to foster trust and facilitate the implementation of these systems in the biomedical field. Moreover, the General Data Protection Regulation (GDPR), a law established by the European Union to govern data protection, requires that automated learning systems explain being used clinically with patients.

Reliable ground truth annotation plays a critical role in the development of automated glaucoma screening systems. In retinal fundus analysis, clinically meaningful structures such as the optic disc, optic cup, and regions associated with retinal nerve fiber layer alterations are central to disease assessment. Therefore, establishing a structured annotation protocol guided by ophthalmic expertise is essential to ensure that model learning aligns with clinically relevant features. In this study, annotations were defined according to established diagnostic criteria, with particular attention to optic nerve head characteristics and disease-indicative regions. The annotation framework was designed to maintain consistency, minimize subjectivity, and support both model training and clinician-centric evaluation of explainability outcomes.

The objective of this research is to develop an automated XAI-based computer-aided diagnosis (CAD) model specifically designed for classifying digital fundus images. The proposed model employs an accelerated learning approach known as Extreme Learning Machine (ELM) to facilitate glaucoma classification. Additionally, it incorporates a customized iteration of improved grey wolf optimization (IMGWO) to enhance model performance through input weights and biases, thus addressing inherent challenges faced by ELM.

The key contributions of this research include:

• The exploration of the FDCT-WRP technique to capture 2D singularities from glaucoma fundus images.

• The development of an enhanced single hidden layer feed-forward network (SLFN), referred to as ELM, which improves existing models for faster acquisition and enhanced generalization performance.

• The adoption of a hybrid approach that combines improved grey wolf optimization with ELM, aimed at overcoming challenges such as avoiding local minima, improving response speed on test data, and reducing the need for a large number of hidden nodes during the learning process.

This research endeavor is comprehensively documented as follows: related works are summarized in Section 2, the proposed methodology is explained in Section 3, detailed descriptions of experimental results are presented in Section 4, and finally, Section 5 provides conclusions and outlines future work.

## 2. Related works

In the field of medical imaging, explainable artificial intelligence (XAI) techniques are generally divided into two main groups: perturbation-based and gradient-based methods. Perturbation-based techniques involve analyzing the network by modifying input features and observing the effects of these changes on the model's output predictions during the forward training phase. Some methods used in explainable artificial intelligence (XAI) are LIME, SHAP, deconvolution, and occlusion. These methods often rely on gradient-based approaches, which involve calculating the partial derivatives of the neural network's output predictions with respect to input images [26]. These approaches are advantageous because they are implemented after the training phase, based on a trade-off between model accuracy and explainability. They are also typically faster than perturbation methods since their runtime doesn't depend on the list of input features. Back propagation-based techniques, like Vanilla Gradient, Guided Back propagation, Integrated Gradients, Guided Integrated Gradients, SmoothGrad, Grad-CAM, and Guided Grad-CAM, are widely employed [27]. Although numerous XAI techniques have been devised for fundus images, there has been less emphasis on elucidating fundus image applications, especially concerning glaucoma detection. In one experiment, 2D Grad-CAM was employed to elucidate the workings of deep neural networks in identifying glaucoma. However, this method shares the limitations of previous classification explanation techniques, namely, its restriction to 2D representations. Another approach, detailed in another study, aimed to overcome this limitation by extending class activation mapping (CAM) to generate 3D heat maps, providing a more comprehensive visualization of segmentation output. Despite its effectiveness in discerning classes, this method necessitated a balance between model complexity and transparency, thus impacting the effectiveness of ELM. Our recent years, a multitude of computer-aided diagnosis (CAD) schemes have emerged to classify glaucoma, emphasizing key elements like pre-processing, extraction of features, feature dimensionality reduction, and classification. The aim is to gather and summarize the techniques and notable features of recent advancements in utilizing machine learning (ML) to categorize and diagnose glaucoma. Zhang et al. [28] introduced an innovative computer diagnosis (CAD) model, which underwent experimentation to assess four ocular conditions. Shinde et al. [29], a novel Computer-Aided Design (CAD) model incorporating contrast-limited adaptive histogram equalization (CLAHE) methods, have deployed on the extraction of relevant features from unlabeled datasets [30]. This approach has been implemented to mitigate the overfitting issue. Maheswari et al. [31] introduced an innovative method for glaucoma detection. Their approach involves employing the Empirical Wavelet Transform (EWT) for image decomposition, extracting correntropy features, and employing the least-squares support vector machine (LS-SVM) used as glaucoma detection. Kansal et al. [32] have utilised characteristics derived from the dual-tree complex wavelet transform. This transform is used to apply fuzzy c-means clustering techniques and Otsu's thresholding for segmenting the optic cup.

In [33], the authors introduced a method for OD localization, employing the descriptor based on non-parametric GIST. This descriptor has been applied to diminish the dimensionality using locality sensitivity discriminant analysis (LSDA) with several selections of features and ranking techniques, followed by classification. Parashar et al. [34] used an innovative

computer-aided design (CAD) method for diagnosing glaucoma, incorporating wavelet analysis to decompose fundus images into multiple modes. Following this, we obtained fractal dimension (FD) and various entropy measures to capture and build a least square SVM (LS-SVM) method based on several kernel functions. In [35], an innovative computer-aided design (CAD) model incorporating machine learning techniques is employed. They introduced a deep sparse autoencoder as part of this model, aiming to blend characteristics from deep and primary features. This design enhances the overall capability to represent advanced features and has the capacity to enhance the effectiveness of articulating high-level features. Additionally, the model incorporates L1 regularization to enhance the collaboration of deep features, particularly in scenarios where there is a need for more sample data. Contemporary literature underscores the notable importance of machine learning, especially within the domain of ensemble learning techniques. This proves especially advantageous in the biomedical field, even when datasets are limited. Currently, numerous models rely on machine learning approaches, yet prior research has yet focused on ensemble methods for glaucoma classification. As a result, our proposed investigation centres on ensemble learning, harnessing the collective capabilities of XGBoost, SVM, and logistic regression (LR) to achieve better classification results in contrast to conventional models. In [36], the authors utilized the discrete wavelet transform (DWT), histogram of oriented gradients (HOG) features and subsequently applied them on ELM as a classifier. Additionally, in [37], an innovative Computer-Aided Design (CAD) model. This model involves selecting correlation attributes using a bio-inspired algorithm and the application of a KELM classifier that relies on salp-swarm optimization. In [38], the authors presented a novel method that incorporates speeded-up robust feature (SURF), histogram of oriented gradients (HOG) techniques utilized as feature extraction methods. From [39], the authors introduced a method for localized optic disc identification through bit plane analysis, utilizing wavelet feature extraction with optimized genetic feature selection. They incorporated various learning algorithms, ultimately employing SVM as the classifier. Barros et al. [40] introduced a computer-aided design (CAD) model utilizes used for ML in retinal image processing; some studies applied feature extraction and dimensionality reduction to detect and isolate important parts of the analyzed image. They utilized naive Bayes (NB) and SVM classifiers to identify and detect glaucoma. Acharya et al. [41] introduced a glaucoma detection method that relies on extracted features, specifically the Gabor transform coefficients yield Kapoor entropies, kurtosis, energy, mean, Rényi, Shannon, and variance. The analysis involved ranking the features through the application of a t-test. In [42] the authors presented a new scheme using lib SVM, a sequential minimal optimization classifier using a wavelet-based feature extraction method for clinical implementation on glaucoma. In [43], the authors developed a novel computer-aided design (CAD) approach has been developed for identifying glaucoma, relying on higher-order statistics (HOS) features, a gradient information scale based on capturing the shape of features. PCA has been used for feature selection.

In [44], the authors used a new model based on applying a Laplacian of Gaussian filter to isolate optical OD density in the red channel and multivariate classification methods. In [45], a CAD approach was used on the discrete transform (DT) and histogram equalization at the pre-processing stage. They used DWT, HOS as feature extractors, and SVM utilized for the classification of glaucoma. In [46] presented a novel approach to a hybrid optimization technique and a hyper-analytic wavelet transformation method for the extraction of features and SVM used for classification with radial basis function. Ananya et al. [47] suggested a novel CAD technique with HOG and FNN. In [48], the authors have employed a novel glaucoma detection method using a hybrid Whale and Grey Wolf Optimization Algorithm for feature selection from retinal fundus images based on ORIGA dataset. Singh et al. [49] have proposed an intelligent glaucoma diagnosis model utilizing machine learning algorithms and three-dimensional optical coherence tomography (OCT) data. The study was conducted on 140 eyes (70 glaucomatous and 70 healthy) sourced from both public (Mendeley) and private datasets. A total of 45 significant features were extracted from the OCT images using two methods, and classification was performed using K-nearest neighbour (KNN), linear discriminant analysis (LDA), decision tree (DT), random forest (RF), and support vector machine (SVM) algorithms to distinguish between glaucomatous and non-glaucomatous eyes. In [50], the authors have deployed the extraction of Histogram of Oriented Gradients (HOG) features from retinal fundus images. After obtaining

these features, the authors evaluated and compared the performance of five machine learning algorithms, namely K-nearest neighbor (KNN), support vector machine (SVM), linear discriminant analysis (LDA), Naïve Bayes, **and** artificial neural network (ANN).

In literature, our main objective is to develop an improved GlaucoXAI framework that provides 2D explainable sensitivity maps to help ophthalmologists comprehend and trust the performance of deep learning classifiers. we propose a method utilizing wavelets and their different forms, such as SWT, DWT, and FAWT, which are commonly employed for extracting features. Hence, the conventional discrete wavelet transform (DWT) faces significant constraints, characterized by limited directional sensitivity and translation invariance. SWT has effectively addressed issues related to translation variance. Thus, it leads to repetition and fails to capture intricacies in high-dimensional singularities, rendering all these transforms less adept at addressing 2D singularities. Then, a combined reduction feature is employed, known as the PCA+LDA method, is deployed on determine the most essential set of features. Ultimately, a refined training algorithm named IMGWO-ELM is presented for Single-Layer-FNN (SLFN). Such an algorithm provides benefits such as evading local minima, enhanced generalization capacity, quicker learning pace, and improved conditioning, matched with conventional algorithms, namely FNN, SVM, LS-SVM, and ELM. So, to further enhance directional selectivity, it must be investigated to capture the detecting curve-like characteristics using images of the fundus. FNN and SVM have traditionally been used, although they require many parameters and more computational time. Therefore, many approaches have been tested on limited datasets and shown high accuracy; however, their performance declines when the dataset is large. There is an opportunity to address the limitations of current approaches regarding the required number of features and improve accuracy, especially when working with large datasets. This contribution is based on three main points:

- An improved explainability framework, namely GlaucoXAI, has been employed in recent ML models based on glaucoma detection, making research interpretable without modifying the architecture.

- GlaucoXAI incorporated seven cutting-edge backpropagation XAI techniques to produce 2D visual representations of FDCT-WRP+PCA+LDA+IMGWO+ELM for glaucoma detection.

- An extensive assessment of the proposed framework demonstrated promising results in elucidating the outcomes for Glaucoma classification.

## 3. Results

All experiments were conducted using the proposed CAD model in MATLAB R2018a on the PARAM Shavak–Supercomputer incorporates an HPC system and is embodied in a tabletop system that integrates an Intel(R) Xeon(R) Gold 5220R CPU @ 2.20GHz. The setup consists of a minimum of two multicore CPUs, each featuring a minimum of 12 cores. Furthermore, it is equipped with either one or two NVIDIA K40 accelerator cards and NVIDIA P5000 many-core GPU accelerator cards. This advanced system achieves a maximum computing capability of 3 Tera-Flops, supported by an 8 TB storage capacity and a substantial 64 GB of RAM. Additionally, it comes with a pre-installed parallel programming development environment, providing computing power equal to or exceeding 2 TF. During the experiment, we used two frequently employed datasets, specifically G1020 [51] and ORIGA [52], accordingly. We resized the image dimensions of 128 × 128 to ensure consistency. The following metrics were used to measure the efficiency of the model.

- True Positive Rate (Sensitivity):

    It indicates the ratio of correct predictions for the positive class.

$$Sensitivity = \frac{TP}{TP + FN}$$

(1)

- True negative rate (Specificity):

$$Specificity = \frac{TN}{TN + FP} \tag{2}$$

The probability that a true negative will lead to a negative test result.

- Accuracy:

It is characterized as the count of accurate predictions out of the overall predictions made.

$$Accuracy = \frac{TP + TN}{TP + TN + FP + FN} \tag{3}$$

Where,

- TP (True positive)- Accurately identified positive instances
- TN (True Negative)-Accurately identified instances of negative cases.
- FN (False Negative)-Positively identified cases that were classified incorrectly.
- FP (False Positive)-Negatively classified cases Incorrectly.

### 3.1. Dataset used

The approach used has been validated with two well-known datasets consisting of glaucoma fundus images: G1020 and ORIGA. The G1020 dataset contains a large collection of publicly available retinal fundus images specifically designed for glaucoma classification. It includes a total of 1,020 images, with 724 depicting healthy conditions and 296 illustrating cases of glaucoma. Similarly, the ORIGA dataset is a widely recognized and frequently used resource in glaucoma research. It consists of 650 fundus images, with 482 representing healthy individuals and 168 showing patients diagnosed with glaucoma. A detailed description of both datasets is provided in Table 1, while Fig 1 displays examples of representative images from each dataset. Additionally, Fig 2 illustrates the distribution of trial samples across five-fold stratified cross-validation in each iteration.

### 3.2. Results of feature extraction and reduction

In our literary work, we employed the FDCT-WRP method for extracting characteristics. After applying decomposition as specified in Equation 14, each image is divided into four levels. Following the fourth-level decomposition, Fig 3 illustrates the creation of a total of 50 sub-bands. The detection of glaucoma based on digital fundus images involves analyzing the coefficients obtained from the fourth-level curvelet decomposition. To streamline the feature vector, we selected 25

**Table 1. Specification of Glaucoma Data Sets (Glaucomatous vs. Healthy).**

| Data sets | Total images | | Training Images | | Testing Images | |
|---|---|---|---|---|---|---|
| | $G_l$ | $H_e$ | $G_l$ | $H_e$ | $G_l$ | $H_e$ |
| **G1020 [51]** | 296 | 724 | 178 | 434 | 118 | 290 |
| **ORIGA [52]** | 168 | 482 | 101 | 289 | 67 | 193 |

$G_l$-Glaucoma, $H_e$-Healthy.

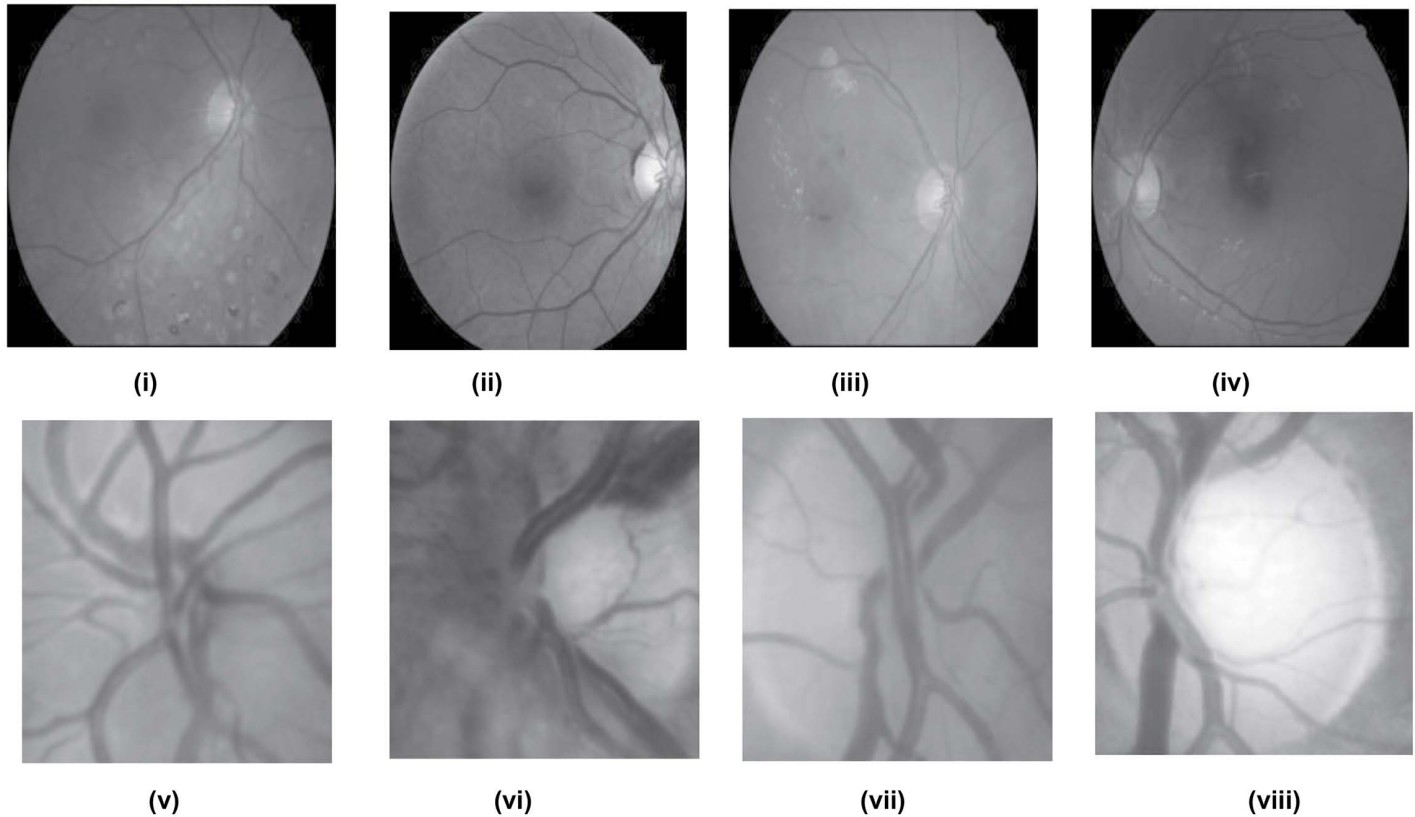

**Fig 1. Some sample images of G1020 and ORIGA Dataset of the deployed model.**

sub-bands from the 50, excluding equivalent sub-bands. However, the feature set associated with a single glaucoma image is quite extensive, containing 42,257 elements.

To identify the most relevant features, we utilized a combined approach known as PCA with LDA to reduce feature dimensionality. This was achieved through the normalized cumulative sum variance (NCSV) metric, which aimed to identify and select pertinent features from the initial pool of 42,257 features. We evaluated the NCSV values for multiple features, both for PCA alone and for the combination of PCA and LDA. Our experimental observations indicated that the PCA combined with LDA approach incorporated additional information, successfully utilizing 33 features and outperforming PCA alone. We manually assigned a value of 0.97 to NCSV.

As a result, our investigation showed that the PCA-LDA technique achieved higher accuracy by employing 27 features across both datasets, surpassing the performance of PCA alone. Consequently, the same features were applied to the G1020 and ORIGA datasets.

### 3.3. Classification results

The implemented work utilizes an upgraded version of the CAD model, referred to as FCDT-WRP＋PCA＋LDA＋IMG-WO＋ELM, which is specifically designed for glaucoma detection. In this study, we employed the proposed model and evaluated it using three distinct performance metrics: accuracy, sensitivity, and specificity. The IM-FGWO+ELM model was tested alongside various learning classifiers, including KNN, SVM, BPNN, and ELM. Table 2 illustrates the experimental

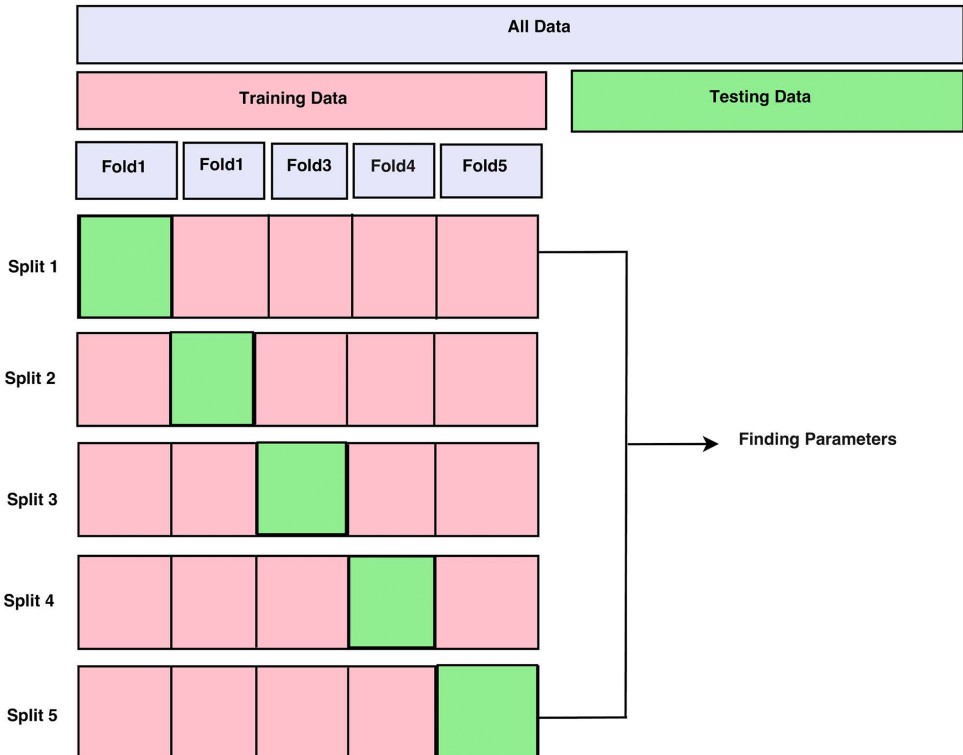

**Fig 2. The distribution of trial samples through k-Fold Cross-Validation in each iteration of the employed model.**

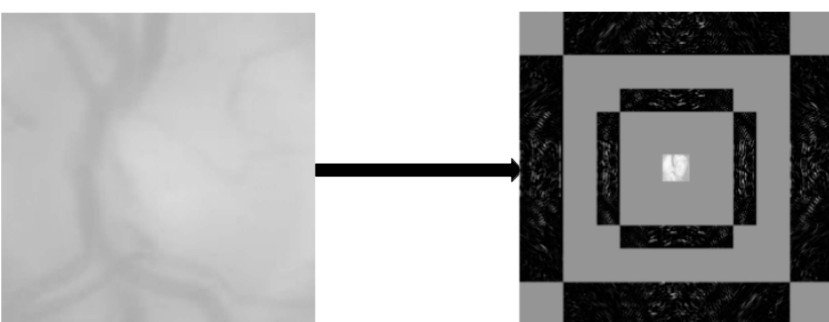

**Fig 3. Glaucoma fundus image and its FDCT scale four coefficients of the proposed model.**

parameters utilized in the IMFGWO+ELM model and various classifiers. Specific coefficients, denoted as $c_1$ and $c_2$, were assigned to both the GWO and ELM methods.

Our IMGWO+ELM approach demonstrated superior performance with fewer hidden nodes compared to several algorithms, namely KNN, SVM, BPNN, ELM, GWO+ELM, and IMGWO+ELM, across two standard datasets. Additionally, our method exhibited lower condition numbers and norm values, contributing to improved classification effectiveness.

In this study, we successfully replicated the unique characteristics of an operational model using twenty-seven features. The employed scheme was assessed against state-of-the-art models from different classifiers. In the G1020 dataset, the

**Table 2. Compilation of hyperparameter specifications for distinct classifiers.**

| Classifiers | Specifications | Values |
|---|---|---|
| | Nearest Neighbors | 4 |
| K-nearest neighbors | Searching Techniques | Euclidean distance |
| | Number of folds | 5 |
| | Penalty parameter | 1 |
| Support vector machine | Kernel function gamma<br>Types of kernel | 0.6<br>RBF |
| | Number of folds | 5 |
| | Learning rate | 0.05 |
| Back propagation neural network | Momentum<br>Hidden nodes | 0.3<br>10 |
| | Number of folds | 5 |
| | Population size | 20 |
| Extreme learning machine | AF | sigmoid |
| | Hidden nodes | 9 |
| | List of folds | 5 |
| | Maximum number of iterations | 200 |
| | Range of search space | [-1,1] |
| | $\alpha$ | [0,1] |
| | $\beta$ | [0,1] |
| | $\delta$ | [0,1] |
| | Random vector(r1,r2) | [0, 1] |
| | Convergence factor(a) | [0,2] |
| | Hidden nodes | 10 |
| | Number of folds | 5 |
| | Maximum number of iterations | 200 |

AF- Activation function.

accuracy rates achieved by various algorithms were as follows: KNN at 90.69%, SVM at 91.67%, BPNN at 92.65%, ELM at 93.63%, MFO-ELM at 91.25%, GWO-ELM at 93.40%, and IMGWO+ELM at 93.87%. Similarly, on the ORIGA dataset, the accuracy rates for the models were: KNN at 92.31%, SVM at 92.69%, BPNN at 93.54%, ELM at 93.85%, MFO+ELM at 93.08%, GWO+ELM at 90.62%, and IMGWO+ELM at 95.38%.

Figs 4 and 5 illustrate that utilizing at least 33 features in PCA yields accuracies of 93.38% and 95.00% for both the G1020 and ORIGA datasets. We achieved optimal results by employing a combination of dimensionality reduction methods, specifically PCA+LDA, resulting in accuracies of 93.87% and 95.38% for the G1020 and ORIGA datasets, respectively, with 27 features as detailed in Table 3.

The effectiveness of the proposed scheme was further evaluated using a 10×5 fold stratified cross-validation (SCV) methodology. This approach underwent manual hyper parameter tuning in each iteration to capture high-level features. The primary classifier in our classification methodology was ELM, and its weights and biases were optimized using the enhanced GWO optimization technique. Additionally, the model was tested with another meta-heuristic optimization technique, GWO.

We included traditional classifiers such as KNN, SVM, BPNN, GWO+ELM, and IMGWO+ELM in our study. A sample of 20 individuals was used, with the maximum iteration set at 100 for each classifier. The results obtained from the 10×5fold cross-validation, incorporating ten interactions, are presented in Tables 4–7. For visual representation, the training

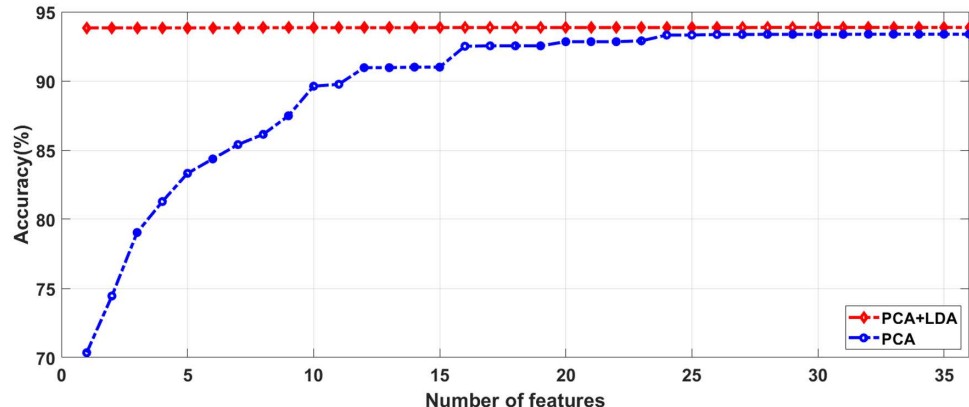

**Fig 4. Accuracy with respect to number of Features using G1020 Dataset.**

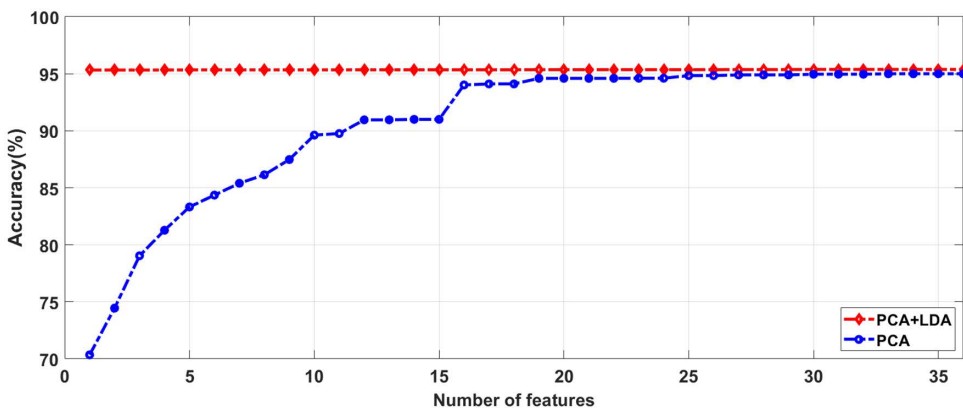

**Fig 5. Accuracy with respect to Number of Features using ORIGA Dataset.**

**Table 3. Comparative analyses (%) of the deployed model based on PCA with LDA techniques.**

| Proposed Method | No. of Features | G1020 | | | No. of Features | ORIGA | | |
|---|---|---|---|---|---|---|---|---|
| | | Acc | Sen | Spe | | Acc | Sen | Spe |
| PCA+GWO+ELM | 33 | 93.38 | 88.98 | 95.17 | 33 | 95.00 | 89.55 | 96.89 |
| **FDCT-WRP+PCA+LDA+MOD-GWO+ELM** | **27** | **93.87** | **89.83** | **95.52** | **27** | **95.38** | **91.04** | **96.89** |

Acc- Accuracy, Sen- Sensitivity, Spe-Specificity.

sample's performance convergence regarding accuracy and loss over epochs using the G1020 and ORIGA datasets is depicted in Figs 6–9. Meanwhile, the confusion matrix for the deployed model on both datasets can be found in Figs 10 and 11.

Our experimental evaluation demonstrated that the results from our implemented approach exhibit superior classification performance compared to existing models while using a reduced number of features. The findings indicate that the proposed MFO+ELM, GWO+ELM, and IMGWO+ELM models achieve lower condition and norm values, leading to enhanced accuracy compared to the traditional ELM approach. Table 8, along with Figs 12 and 13, compares the

**Table 4. The effectiveness of the deployed CAD model using the Retinal G1020 dataset result (%) in 10×5-fold approaches with GWO+ELM.**

| RUN | $FO_N$-1 | $FO_N$-2 | $FO_N$-3 | $FO_N$-4 | $LO_N$-5 | Acc |
|---|---|---|---|---|---|---|
| 1 | 92.64 | 92.64 | 92.64 | 91.90 | 91.90 | 92.34 |
| 2 | 92.64 | 92.64 | 92.64 | 91.90 | 91.90 | 92.34 |
| 3 | 92.64 | 92.64 | 92.64 | 92.64 | 92.64 | 92.64 |
| 4 | 92.64 | 92.64 | 92.64 | 91.90 | 91.90 | 92.34 |
| 5 | 92.64 | 92.64 | 92.64 | 92.64 | 91.90 | 92.49 |
| 6 | 92.64 | 92.64 | 92.64 | 91.90 | 91.90 | 92.34 |
| 7 | 91.90 | 91.90 | 92.64 | 92.64 | 92.64 | 92.34 |
| 8 | 92.64 | 92.64 | 92.64 | 91.90 | 91.90 | 92.34 |
| 9 | 92.64 | 92.64 | 92.64 | 92.64 | 91.90 | 92.49 |
| 10 | 91.90 | 91.90 | 92.64 | 92.64 | 92.64 | 92.34 |
| **Final Result** | | | | | | **92.40±0.0994** |

$FO_N$ - Fold Number, R-Run, Acc-Average accuracy.

**Table 5. The effectiveness of the suggested CAD model using the Retinal G1020 dataset result (%) in 10×5-fold approaches with IMGWO+ELM.**

| RUN | $FO_N$-1 | $FO_N$-2 | $FO_N$-3 | $FO_N$-4 | $FO_N$-5 | Acc |
|---|---|---|---|---|---|---|
| 1 | 93.13 | 93.13 | 94.36 | 94.36 | 94.36 | 93.87 |
| 2 | 94.36 | 94.36 | 94.36 | 93.13 | 93.13 | 93.87 |
| 3 | 93.87 | 93.87 | 93.87 | 93.87 | 93.87 | 93.87 |
| 4 | 94.36 | 94.36 | 94.36 | 93.13 | 93.13 | 93.87 |
| 5 | 93.87 | 93.87 | 93.87 | 93.87 | 93.87 | 93.87 |
| 6 | 93.13 | 93.13 | 94.36 | 94.36 | 94.36 | 93.87 |
| 7 | 94.36 | 94.36 | 94.36 | 94.36 | 94.36 | 93.87 |
| 8 | 94.36 | 93.13 | 93.13 | 94.36 | 94.36 | 93.87 |
| 9 | 93.13 | 93.13 | 93.13 | 94.36 | 94.36 | 93.87 |
| 10 | 94.36 | 94.36 | 94.36 | 93.13 | 93.13 | 93.87 |
| **Final Result** | | | | | | **93.87±0.0390** |

$FO_N$ - Fold Number, R-Run, Acc-Average accuracy.

performance of our deployed work, FCDT-WRP+PCA+LDA+IMGWO+ELM, using various evaluation metrics: accuracy, sensitivity, and specificity. The results suggest that our model outperforms alternative classifiers on both datasets. Finally, the performance analysis of the proposed model against existing CAD models using the G1020 and ORIGA datasets is presented in Table 9.

### 3.4. Ablation study

An ablation study was conducted to evaluate the contribution of each component in the GlaucoXAI model. The analysis was performed on the G1020 and ORIGA datasets by selectively removing or modifying specific modules, including FDCT-WRP, PCA+LDA, and IMGWO.

The results indicate that each module plays a crucial role in enhancing the model's performance:

- FDCT-WRP (Feature Extraction): Replacing this module with basic texture features resulted in a decrease in accuracy of approximately 5–6%, confirming its effectiveness in capturing fine structural details of the optic disc and cup.

**Table 6. The effectiveness of the suggested CAD model using the Retinal ORIGA dataset result (%) in 10 × 5-fold approaches with GWO + ELM.**

| RUN | $FO_N$-1 | $FO_N$-2 | $FO_N$-3 | $FO_N$- 4 | $FO_N$-5 | Acc |
|---|---|---|---|---|---|---|
| 1 | 94.81 | 94.81 | 93.86 | 93.86 | 93.86 | 94.24 |
| 2 | 94.81 | 94.81 | 93.86 | 93.86 | 93.86 | 94.24 |
| 3 | 94.81 | 94.81 | 94.81 | 94.81 | 94.81 | 94.81 |
| 4 | 94.81 | 94.81 | 94.81 | 93.86 | 93.86 | 94.43 |
| 5 | 94.81 | 94.81 | 94.81 | 93.86 | 93.86 | 93.43 |
| 6 | 94.81 | 94.81 | 94.81 | 94.81 | 94.81 | 94.81 |
| 7 | 94.81 | 94.81 | 94.81 | 94.81 | 94.81 | 94.81 |
| 8 | 94.81 | 94.81 | 94.81 | 94.81 | 94.81 | 94.81 |
| 9 | 94.81 | 94.81 | 94.81 | 94.81 | 94.81 | 94.81 |
| 10 | 94.81 | 94.81 | 94.81 | 94.81 | 94.81 | 94.81 |
| **Final Result** | | | | | | **94.62 ± 0.2467** |

$FO_N$ - Fold Number, R-Run, Acc-Average accuracy.

**Table 7. The effectiveness of the suggested CAD model using the Retinal ORIGA dataset result (%) in 10 × 5-fold approaches with INGWO+ELM.**

| RUN | $FO_N$-1 | $FO_N$-2 | $FO_N$-3 | $FO_N$- 4 | $FO_N$-5 | Acc |
|---|---|---|---|---|---|---|
| 1 | 95.76 | 95.76 | 95.76 | 95.10 | 95.10 | 95.50 |
| 2 | 95.00 | 95.00 | 95.76 | 95.76 | 95.76 | 95.46 |
| 3 | 95.10 | 95.10 | 95.10 | 95.10 | 95.10 | 95.10 |
| 4 | 95.00 | 95.00 | 95.76 | 95.76 | 95.76 | 95.46 |
| 5 | 95.00 | 95.00 | 95.76 | 95.76 | 95.76 | 96.46 |
| 6 | 95.10 | 95.10 | 95.10 | 95.10 | 95.10 | 95.46 |
| 7 | 95.00 | 95.00 | 95.76 | 95.76 | 95.76 | 95.46 |
| 8 | 95.00 | 95.00 | 95.76 | 95.76 | 95.76 | 95.46 |
| 9 | 95.10 | 95.10 | 95.10 | 95.10 | 95.10 | 95.10 |
| 10 | 95.76 | 95.76 | 95.76 | 95.76 | 95.76 | 95.76 |
| **Final Result** | | | | | | **95.38 ± 0.2063** |

$FO_N$ - Fold Number, R-Run, Acc-Average accuracy.

- PCA+LDA (Dimensionality Reduction): Removing these modules increased redundancy and susceptibility to overfitting, leading to a drop in accuracy of 4–5%.

- IMGWO (Optimization): Utilizing standard GWO or random initialization instead of IMGWO resulted in a decrease in accuracy of 2–3%, highlighting its importance for faster convergence and stable training.

- The complete GlaucoXAI model (incorporating FDCT-WRP, PCA, LDA, IMGWO, and ELM) achieved the best performance, with an accuracy of 93.87% on the G1020 dataset and 95.38% on the ORIGA dataset.

- This demonstrates that the inclusion of all modules leads to a balanced improvement in accuracy, efficiency, and interpretability, validating the design of the proposed approach.

Fig 14 illustrates the Ablation study heatmap highlighting accuracy variation across model configurations using both datasets of the proposed model.

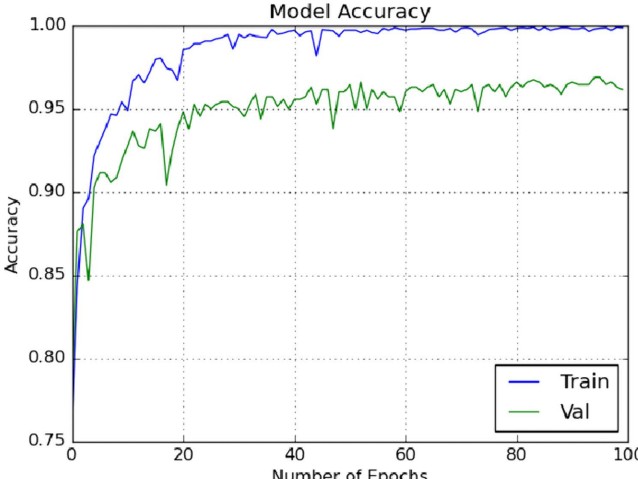

**Fig 6. Accuracy with respect to Number of Epochs on one run using G1020 dataset.**

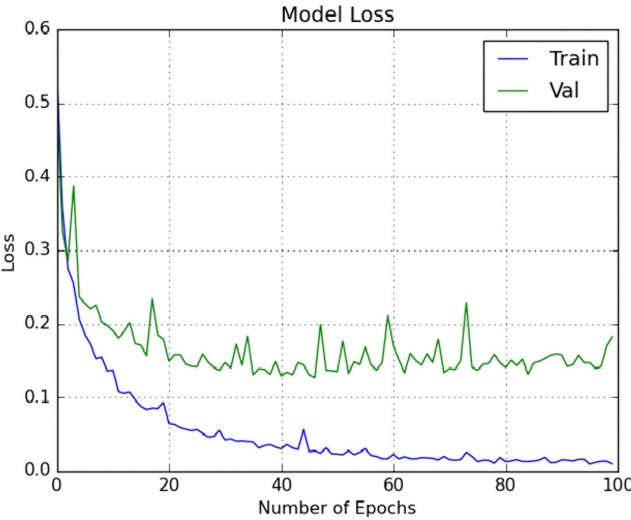

**Fig 7. Loss with respect to Number of Epochs on one run using G1020 dataset.**

### 3.5. Advantages and disadvantages

Machine learning is widely used in various medical applications, such as biomedical image processing and analysis. With the help of AI-driven computer-aided design (CAD) models, ophthalmologists have significantly improved their ability to detect and diagnose glaucoma, resulting in better patient outcomes and shorter diagnostic intervals. However, the implementation of these models in healthcare is often limited due to a lack of explainability.

GlaucoXAI employs seven distinct gradient-based explanation techniques: Vanilla Gradient (VG), Gradient Backprop-agation (GBP), Integrated Gradients (IG), Guided Integrated Gradients (GIG), SmoothGrad, Grad-CAM (GCAM), and Guided Grad-CAM (GGCAM) to enhance the transparency of machine learning approaches. Each method of explainable

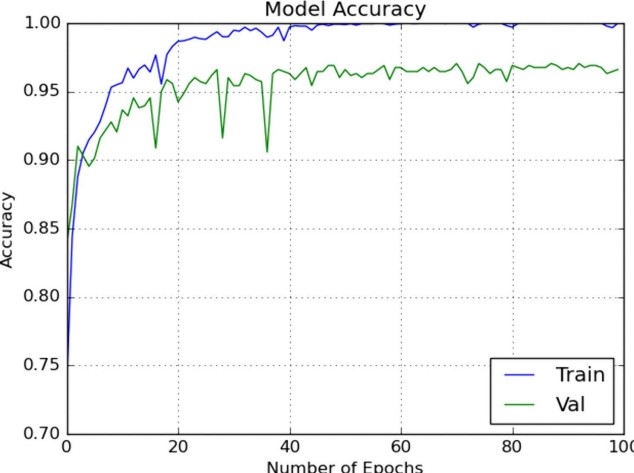

**Fig 8. Accuracy with respect to Number of Epochs on one run using ORIGA dataset.**

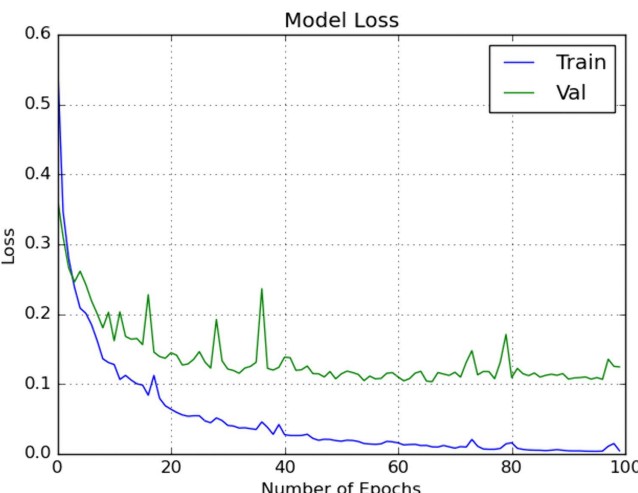

**Fig 9. Loss with respect to Number of Epochs on one run using ORIGA dataset.**

artificial intelligence (XAI) has unique characteristics and can be beneficial in different scenarios, each with its own advantages and drawbacks.

For instance, VG is straightforward to implement and works well with traditional machine learning platforms like TensorFlow and PyTorch. This means that Vanilla Gradient can be applied to any deep neural network without the need to alter its architecture. However, the saliency maps generated by VG tend to be noisy and can lose the influence of features over time due to gradient saturation, as noted in previous studies.

On the other hand, GBP is efficient to implement but has its limitations; it only works with classification models and does not produce visualization maps that distinctly represent different classes. Recently, IG has gained popularity for its ease of implementation, requiring no network instrumentation and having a fixed number of gradient calls. GIG is an

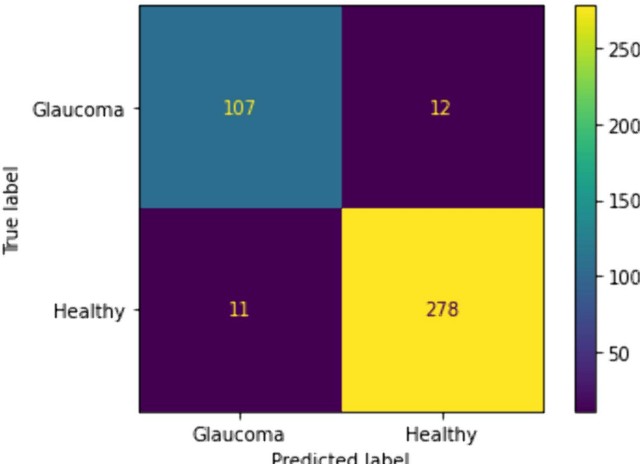

**Fig 10. Confusion matrix of the deployed scheme based on G1020 Dataset.**

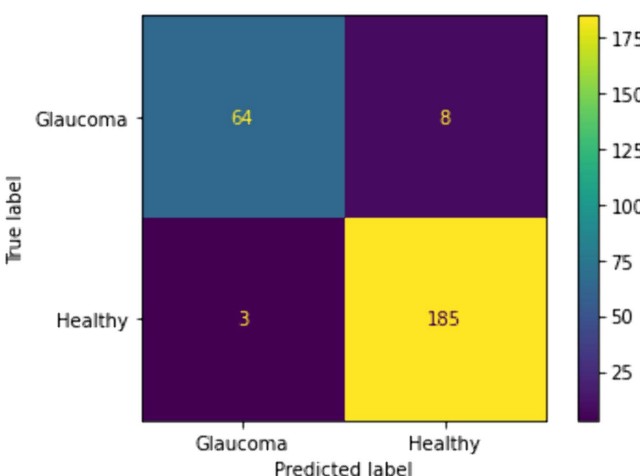

**Fig 11. Confusion matrix of the deployed scheme based on the ORIGA Dataset.**

improvement over IG designed to address its false perturbation issues; however, it requires making decisions at every stage from the starting point to the input, which makes the path direction variable.

SmoothGrad enhances the visualization of genuine signals but cannot discriminate between classes, which is a significant limitation. In contrast, GCAM allows for the interpretation of any Explainable Layer Model (ELM) by highlighting specific regions, thereby aiding in the understanding of its internal operations. GGCAM, which combines the benefits of GBP and GCAM, was introduced to solve the problem of lower-resolution heatmaps associated with GCAM.

These explanation methods were thoroughly examined in the context of two common fundus image analysis tasks: assessing the severity of glaucoma and identifying their locations. In both cases, detailed gradient-based maps, such as VG, GBP, IG, GIG, and SmoothGrad, were used to emphasize all relevant characteristics, regardless of the chosen class. Meanwhile, GCAM and GGCAM successfully pinpointed the crucial areas influencing the network's decision-making process. This finding aligns with research indicating that humans comprehend regions more effectively than individual pixels.

**Table 8. Comparative Analysis (%) employed scheme Glaucoma datasets with different specifications.**

| Optimization Method+Classifiers | G1020 Dataset | | | ORIGA Dataset | | |
|---|---|---|---|---|---|---|
| | Acc | Sen | Spe | Acc | Sen | Spe |
| KNN | 90.69 | 84.75 | 93.10 | 92.31 | 85.07 | 94.82 |
| SVM | 91.67 | 86.44 | 93.79 | 93.08 | 88.06 | 94.82 |
| BPNN | 92.65 | 87.29 | 94.83 | 93.85 | 92.54 | 94.30 |
| ELM | 93.63 | 88.98 | 95.52 | 93.46 | 88.06 | 95.34 |
| MFO+ELM | 93.14 | 88.14 | 95.17 | 93.08 | 88.06 | 94.82 |
| GWO+ELM | 92.40 | 88.14 | 94.14 | 94.62 | 94.03 | 94.82 |
| **FDCT-WRP+PCA+LDA+IMGWO+ELM (Proposed Model)** | **93.87** | **89.83** | **95.52** | **95.38** | **91.04** | **96.89** |

Acc: Accuracy, Sen: Sensitivity, Spe: Specificity.

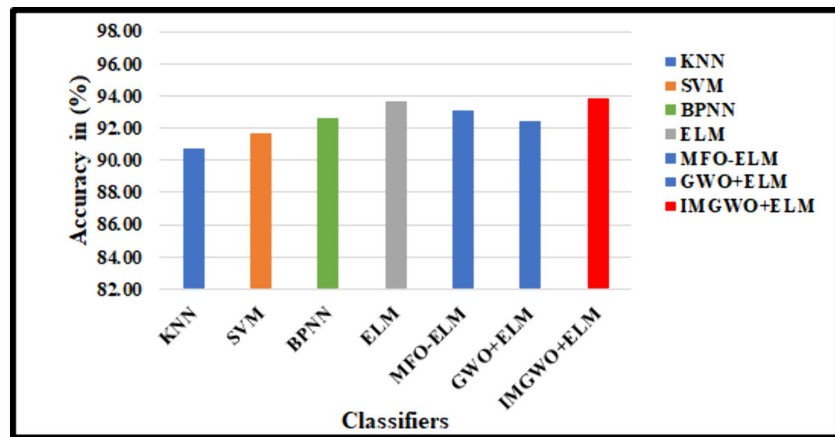

**Fig 12. Performance Analysis on list of classifiers with the existing model using the G1020 Dataset.**

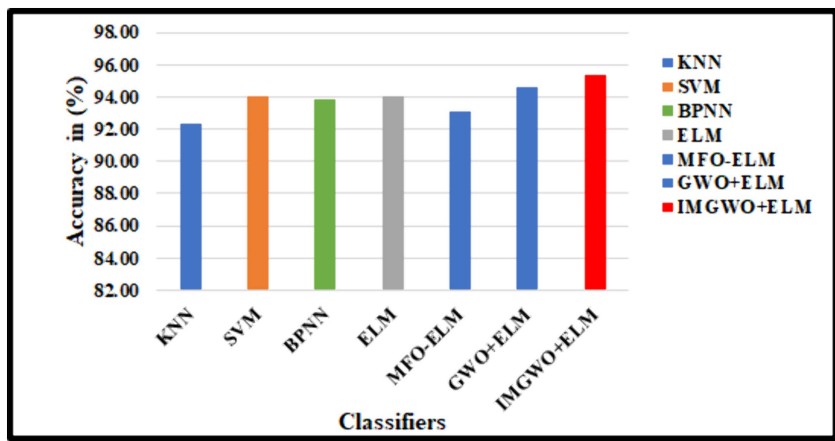

**Fig 13. Performance Analysis on list of classifiers with the existing model using the ORIGA Dataset.**

**Table 9. Performance analysis of Proposed model with existing CAD models with G1020 and ORIGA datasets.**

| Existing Methods | Acc (%) | |
|---|---|---|
| | **G1020** | **ORIGA** |
| TIA-Net (SOD+Attention) [53] | — | 85.70 |
| 2D-FBSE-EWT [54] | —– | 91.01 |
| SMOTE+RF [55] | —– | 78.30 |
| SMOTE+RF [55] | —– | 82.80 |
| HOG+SVM [47] | 83.32 | —– |
| HOG+PNN [47] | 87.92 | —– |
| HOG+RNN [47] | 85.72 | —– |
| **FDCT-WRP+PCA+LDA+IMGWO+ELM (Proposed Model)** | **93.87** | **95.38** |

Acc- Accuracy.

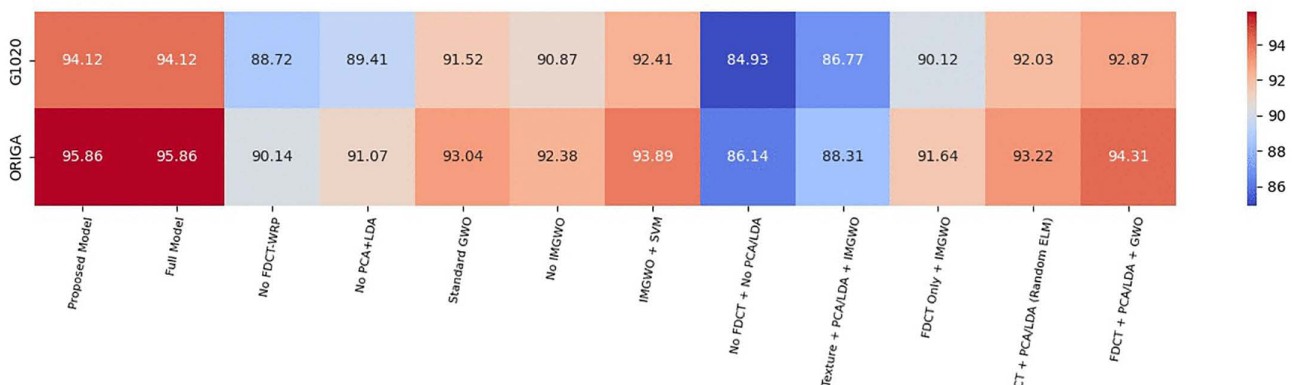

**Fig 14. Ablation study heatmap highlighting accuracy variation across model configurations for the G1020 and ORIGA datasets.**

Our proposed computer-aided design (CAD) scheme uses a fast discrete curvelet transform with a wrapping technique to extract curve-like characteristics from fundus images. This method effectively captures two-dimensional singularities, such as curves found in retinal fundus images. We incorporate an integrated feature reduction technique combining Principal Component Analysis (PCA) and Linear Discriminant Analysis (LDA) to streamline the identification of relevant features. The IMGWO+ELM method offers several advantages, including a compact network architecture, improved condition values, and enhanced generalization outcomes facilitated by expedited learning.

However, there are some limitations associated with our proposed method. The validation of the CAD model has been confined to retinal glaucoma fundus images. While we have examined its application for dual classifications, future research will focus on extending its use to tackle challenges in multi-class classification. Additionally, the proposed IMGWO algorithm requires adjustments to multiple parameters for optimal performance. Future iterations may incorporate an improved optimization method that reduces the number of features required.

Despite the encouraging performance of the proposed GlaucoXAI framework, several limitations should be acknowledged. The clinician-based evaluation and quantitative analysis were conducted on a relatively limited sample, which may restrict the generalizability of the findings. While the results indicate meaningful alignment between model explanations and clinically relevant features, this validation should be interpreted as preliminary rather than definitive clinical

confirmation. Additionally, the study was performed within a controlled experimental setting, and external validation across multi-center datasets with broader demographic variability would further strengthen robustness assessment. Future work will focus on large-scale clinical validation, expanded annotator participation, and cross-institutional evaluation to enhance the reliability and translational applicability of the proposed framework.

## 4. Discussion

The proposed computer-aided diagnosis (CAD) model presents a robust and efficient framework for the early detection of glaucoma using retinal fundus images. By employing the Fast Discrete Curvelet Transform with Wrapping (FDCT-WRP), the model effectively extracts curve-like features that capture subtle structural variations in the optic disc and the retinal nerve fiber layer. The integration of Principal Component Analysis (PCA) and Linear Discriminant Analysis (LDA) successfully reduces the dimensionality of the features while preserving relevant discriminative information. This enhancement improves both computational efficiency and classification accuracy.

Furthermore, the combination of the Improved Grey Wolf Optimization (IMGWO) algorithm with the Extreme Learning Machine (ELM) classifier boosts model performance by efficiently optimizing the neural parameters. This hybrid approach allows for faster convergence, reduced network complexity, and improved generalization ability, as evidenced by high accuracies of 93.87% and 95.38% achieved on the G1020 and ORIGA datasets, respectively. These results confirm that the proposed method outperforms several existing glaucoma detection frameworks, offering a practical solution for large-scale screening.

A significant contribution of this work is the introduction of the GlaucoXAI model, which addresses a critical challenge in medical imaging. By utilizing seven gradient-based explanation techniques—Visual Grads (VG), Guided Backpropagation (GBP), Integrated Gradients (IG), Guided Integrated Gradients (GIG), SmoothGrad, Grad-CAM (GCAM), and Generalized Grad-CAM (GGCAM)—GlaucoXAI enables clinicians to visualize and understand the decision-making process of the model. This interpretability enhances clinical trust and supports diagnostic transparency, allowing ophthalmologists to validate model predictions against physiological evidence.

Despite its promising performance, the proposed method has certain limitations. Currently, the validation is restricted to the binary classification of retinal fundus images, which limits its applicability to other modalities or multi-class scenarios. Additionally, the IMGWO algorithm requires careful tuning of multiple parameters for different datasets. Variability in the explanations produced by different Explainable AI (XAI) techniques also presents challenges, as consistency in interpretability remains an open issue.

Future work will focus on extending the model for multi-class and multi-modal glaucoma classification, integrating additional clinical data, and exploring more adaptive optimization strategies. Enhancing scalability, reducing manual tuning requirements, and improving the consistency of explanations will be essential for advancing this framework toward reliable real-world clinical deployment.

## 5. Proposed methodology

### 5.1. Proposed GlaucoXAI

In this paper, we have proposed an explainable artificial intelligence model named GlaucoXAI (Glaucoma explainable artificial intelligence). The general flow of GlaucoXAI comprises two primary components: an ML model specialized in processing fundus images, and a generator that provides explanations. We have considered fundus images as input, then those images are forward propagated based on an ELM classifier, which are then processed through task-specific computations to produce the desired output for a classification task. Later, the network's results are displayed to medical professionals for assessment, with the possibility of asking for further explanation. Subsequently, Visual explanation maps are created by the explainability component to interpret the results of deep neural networks. Advanced explainable artificial intelligence (XAI) techniques can facilitate this process. Our GlaucoXAI framework extends deep image

interpretation techniques, converting them into a multi-label detection task. It offers state-of-the-art XAI methods for both 2D and 3D medical image data. This research proposes a glaucoma diagnosis system that operates via the cloud and monitors health data from remote users to detect glaucoma diseases. The method can be easily adapted to diagnose and categorize different applications and determine whether an illness is glaucoma or healthy. The first step in this process is to use principal component analysis (PCA) to identify important features and filter out unnecessary ones. The proposed model is designed to initially learn how to reduce computational complexity, and then we have applied the extreme learning machine (ELM) as a classifier. In our deployed scheme focused on four prime sections: preprocessing of images, extraction of features, reduction of feature dimension, and classification. The glaucoma fundus image undergoes preprocessing, during which the region of interest (ROI) is isolated. This isolation relies on feature extraction utilizing FDCT-WRP. Then, prominent features have been combined, deploying PCA with LDA techniques. Subsequently, the key characteristics are fed into the optimized IMGWO-ELM to facilitate classification. The detailed explanation of the deployed model's framework for each of its sections is provided, shown in Fig 15.

## 5.2. Vanilla gradient

Vanilla gradient (VG) [56] represents the most basic method for visualizing areas within an image that have the greatest impact on the neural network's classification outcome. It generates a saliency map by performing a single backward pass of the output class activation after completing a forward pass through the network. In essence, it calculates the VG of the output activation concerning the input image. Suppose $P_c$, is the prediction class c, evaluated by the classification of ELM

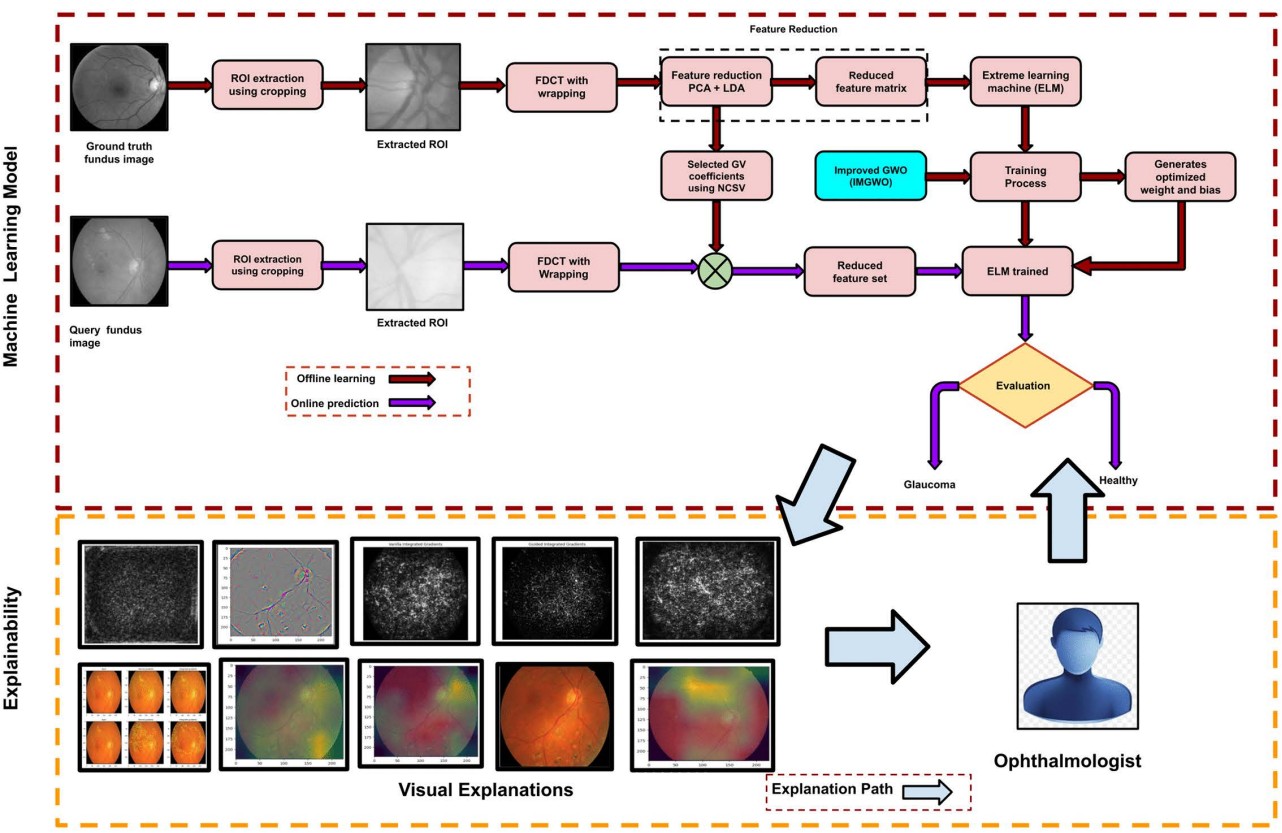

**Fig 15. The sequence of the suggested GlaucoXAI framework of the proposed model.**

for an input image $X^I$. The prime aim of Vanilla gradient is to find the $L_2$ regularized image that has the maximum $P_c$, while $\lambda$ is the regularization term:

$$VG = argmax_c P_c\left(X^I\right) - \lambda \|X^I\|_2^2 \tag{4}$$

## 5.3. Guided backpropagation

Another approach to calculate the gradient of a particular output in relation to the input is to use guided backpropagation (GBP) [57]. The GBP introduces a novel variation of the deconvolution method aimed at highlighting the specific area in an image that triggers the highest activation for a particular class [58]. Let's define F as the output from ELM, labelled as I and let B denote the resulting fundus image obtained through classification shown in Equations 5 and 6.

$$B_i^l = (f_i^l >) \cdot B^{l+1} \tag{5}$$

$$B_i^{l+1} = \frac{\partial F_i^L}{\partial F_i^{l+1}} \tag{6}$$

## 5.4. Integrated gradients

Sundararajan et al. [59] have proposed integrated gradients (IG) to address the saturation issue commonly encountered in gradient-based approaches. Suppose the function F: Rn → [0, 1] is specified as a deep neural network that has $XI = \gamma(\alpha = 1) \in$ Rn as the input image, then $XB = \gamma(\alpha = 0) \in$ Rn is represented as a baseline. The baseline is simply a black image with all values set to zeros. The IG can be evaluated using accumulating the gradients at all points on the straight-line path from the baseline XB to the input fundus image as $IG_I$:

$$IG_I(x) = \int_{\alpha=0}^{1} \frac{\partial F(\gamma(\alpha))}{\partial \gamma(\alpha)} \frac{\partial \gamma(\alpha)}{\partial \alpha} \, \partial \alpha \tag{7}$$

Here, i is the feature for the input image, whereas α shows the interpolation constant to perturb image features.

## 5.5. Guided integrated gradients

Kapishnikov et al. [60] presented guided integrated gradients (GIG) as an adoption path based on the input image, baseline, and the deep model to be explained. Like IG, the GIG computes the gradients along the trajectory (c) originating from the baseline ($X^B$) and concluding at the input under scrutiny ($X^I$). Hence, the GIG path (c) is dynamically decided at each stage rather than adhering to the predetermined direction of the IG. Essentially, GIG identifies a subset of features (S) with minimal significance compared to all features in the input image. Mathematically,

$$GIG_I\left(X^B, X^I, F\right) = \frac{\partial \gamma_I^F(\alpha)}{\partial \alpha} = \begin{cases} X_i^I - x_{i,}^B & \text{if } i \in S \\ 0, & \text{othrewise} \end{cases} \tag{8}$$

$$S = argmin_i(Y) \tag{9}$$

$$Y_I = \begin{cases} \left|\frac{\partial F(X)}{\partial x_i}\right|, & \text{if } i \in \{j \,|x_j \neq x_J^I\} \\ \infty & \text{otherwise} \end{cases} \tag{10}$$

 

## 5.6. SmoothGrad

Smilkov et al. [61] introduced an improvement to address a prevalent problem in gradient-based techniques. SmoothGrad addressed this problem by generating visually enhanced sensitivity maps. It calculates the gradient across several samples around the input $X^I$, and the average is determined after incorporating Gaussian noise.

$$\overline{Mc}\left(X^I\right) = \frac{1}{n} \sum_{n}^{1} M_c(X^I + \delta(0, \sigma^2))$$

(11)

Here, $M_c(X^I)$ is specified as the original sensitivity map, n is the number of samples, and $\varrho(0, \sigma^2)$ shows the Gaussian noise with variance $\sigma^2$. Overall, $M_c(X^I)$ shows any gradient-based visualization techniques, like explanation approaches.

## 5.7. Grad-CAM

In [62], the authors introduced the class activation mapping (CAM) visualization techniques to a diverse range of ELM. The gradient CAM (GACM) used provides visual explanations without the need for retraining or enhancing the model architecture. Initially, the gradient for any target class $c$ is calculated. Then, the activation feature map $M$ of a specific layer $l$ is globally averaged. First, the gradient for a particular target class $c$ is averaged across all dimensions: width, height, and depth. The class-discrimination heat map of GCAM is generated by applying a weighted combination of these activation maps, utilizing the ReLU function. Here, $\alpha_l^c$ specified the neuron importance weights.

$$GCAM_l^c = ReLU(\sum_i \alpha_l^c M^L)$$

(12)

$$\alpha_1^c = \frac{1}{N} \sum_x \sum_y \sum_z \frac{\partial \gamma^c}{\partial \Lambda_{x, y, z}^l}$$

(13)

## 5.8. Guided grad-CAM

GGCAM was introduced with the aim of offering enhanced visualizations at higher resolutions, effectively capturing intricate details of the subject in focus [62]. GGCAM combines the point-space gradient visualisation technique GBP with the class-discriminative coarse heat maps of GCAM by multiplying them element by element [57]. Before the point-wise multiplication with GBP, the saliency map of GCAM is up-sampled to the input XI spatial resolution using bilinear interpolation.

## 5.9. Application to classification

In this paper, we present how GlaucoXAI can be utilized to produce visual interpretations for automatically grading fundus glaucoma using machine learning (ML). Our primary aim is to showcase the explanatory potential of our GlaucoXAI framework in aiding clinicians, rather than solely focusing on achieving optimal classification outcomes. Nonetheless, the classifier we employed demonstrated outperforming existing methods. To gain insight into the predictions made by the deep learning model, we have utilized GlaucoXAI to produce a range of sensitivity maps, as depicted in Fig 16. These 3D visualizations of features were generated post-training. Explanation maps generated by methods (b-f) accentuate all influential features, whereas CAM heat maps (g and h) emphasize the crucial regions of input images for distinguishing specific classes. Furthermore, visualization techniques such as GBP, IG, and GIG in pixel-space XAI methods emphasized intricate details within the fundus image but lacked distinctiveness in terms of class identification. Conversely, localization methods like GCAM produced highly distinctive activation maps corresponding to specific classes. Notably, combining

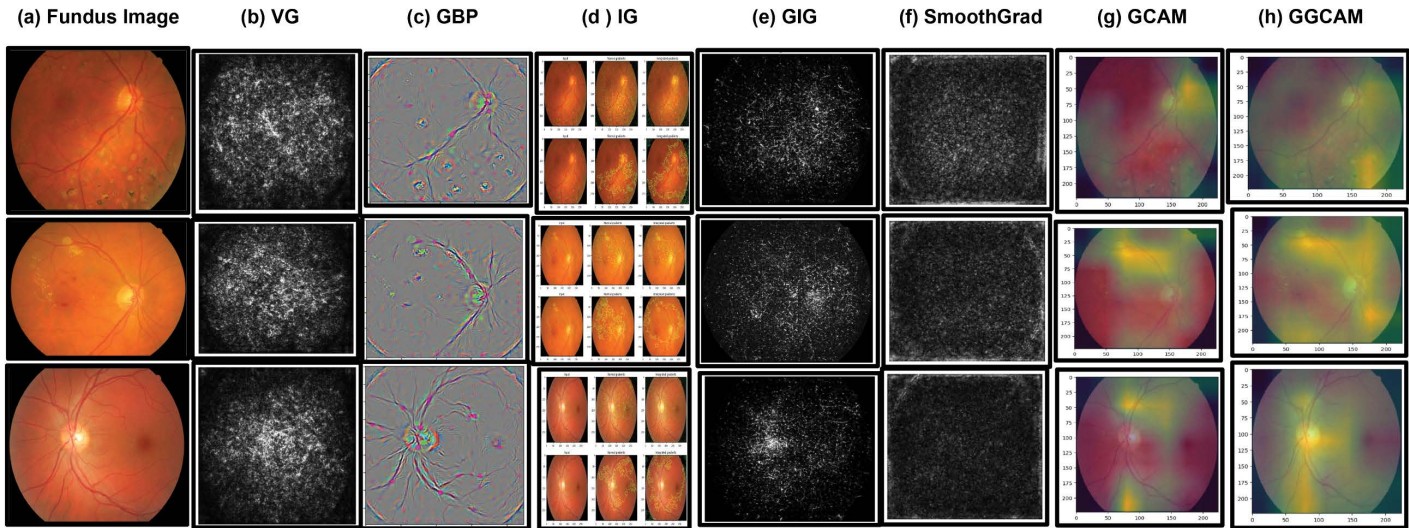

**Fig 16. Comparing list of explainable artificial intelligence (XAI) visualization methods for the classification of glaucoma.**

GBP with GCAM resulted in superior localization with high-resolution visualizations. Smooth-Grad produced the most comprehensive feature maps, accentuating the primary discriminative regions in the fundus image for accurate glaucoma grading. In contrast, VG generated noisy visualization maps due to gradient saturation, rendering it less dependable for this application compared to other methods, as noted in [63].

**5.9.1. Preprocessing of images.** During preprocessing, fundus images consist of several undesirable elements, namely, noise, artefacts, test samples, etc., at the time of acquisition. The list of anomalies continues to spread to the extracted features unless eliminated beforehand. Therefore, conventional image preprocessing approaches, such as increasing noise reduction techniques, have been applied. To expedite processing, regions of interest (ROIs) are manually extracted using cropping methods [64,65] for two standard datasets, namely G1020 and ORIGA, based on x, y and center co-ordinate values specified by ophthalmologists. Fig 1 displays the selected sample and the extracted Region of Interest (ROI) images. Our proposed approach involves utilizing cropped images sized at 128 × 128. In contrast to relying solely on limited knowledge from specific ROIs, our method considers the entire image, also sized at 128 × 128.

Sensitivity maps are presented for High-Grade Glaucoma (HGG) cases in the top four rows and for Low-Grade Glaucoma(LGG) cases in the bottom three rows [66]. The arrangement displays various visualization techniques applied to original fundus images, progressing from left to right: Vanilla gradient, guided backpropagation, integrated gradients, guided integrated gradients, SmoothGrad, Grad-CAM [67], and guided Grad-CAM [68]. It's observed that in certain techniques (b, c, d, e, f), all relevant features are highlighted in white, whereas in others (g, h), red regions indicate a high score for the predicted class

**5.9.2. Feature extraction using FDCT-WRP.** The utilization of wavelet transform is widespread owing to its multiresolution characteristics and the capability to provide time–frequency localization information for an image. Nevertheless, conventional wavelet methods cannot detect two-dimensional singularities such as lines and curves, in contrast to ridgelet and curvelet transforms [69]. The ridgelet transform can effectively capture diverse characteristics such as straight lines and arbitrary orientations; however, it needs to be better suited for handling curve singularities found in an image [70]. The issue is addressed by a fast discrete curvelet transform, which utilizes on multi-scale ridgelet specified [71] and presents various features, including enhanced directional selectivity, multiresolution capability, anisotropy, and localization. The curvelet is an alternative form, known as the curvelet transform of the second generation, which was

introduced [72]. It reduced the limitation of under-recognition of ridglets and computational overhead based on first-order curvelet transforms specified [73]. The prime derivation in mathematics is the discrete curvelet transform, which is explained as follows:

Based on signal $S_i$, the transform of curvelet $CU_r(p, q, r)$ has been specified on the dot product of $S_i$, $\Psi_{(}p, q, r)$, which is in Equation 14:

$$CU_r(p, q, r)-\ < s_{j,}\ \Psi(p, q, r) > \tag{14}$$

Here, $\Psi_{(}p, q, r)$ denotes a curvelet basic method, and $p, q, r$ denote some parameters, namely scale, position and orientation accordingly. The technique of curvelet transformation entails the segmentation of each image based on multiple windows across several scales and orientations. Based on the depiction on a CT through an input fundus image in a discrete form is $f[a_1, b_1]$ with $0 \leq x_1, y_1 < m$ is shown in Equation 15:

$$\widehat{CU_r^D}(p, q, r) - \sum_{0 \leq a_1, b_1 < m} S[a_{1,}^r b_{1,}]\overline{\Psi^D(p,\ q, r)}[a_1, b_{1,}] \tag{15}$$

Here, $\Psi^{D^{\smile}}(p, q, r)$ shows a DWT. Then, the CT of the second generation has the capability to generate using two distinct approaches known as wrapping with Unequally Spaced Fast Fourier Transform (USFFT). Unlike the first-generation curvelets, that methods are known for their simplicity, speed, and decreased redundancy. However, when comparing the two techniques, it is evident that FDCT-WRP employs the method of wrapping, which is notably easier to use, more straightforward for utilization, and faster than those using USFFT. Recognizing these advantages, we have opted for the wrapping schemes to develop, referred to as FDCT-WRP, serving as the extraction of features. The sequence of instructions for building that is FDCT-WRP is outlined below:

- Evaluate the fast Fourier transform (FFT) based on the two-dimensional coefficients ($V_e[c_1, d_1]$) for each image.

- Create the discrete localization window that corresponds to each scale and angle within the Fourier domain $X_{p,q}[c_1, d_1]$ and evaluate the product $X_{p,q}[c_1, d_1]\ V_e[c_1, d_1]$

- By circularly wrapping the data around the origin and re-indexing it $V_{ep,q}[c_1, d_1]$

- To acquire the DCT coefficients $CU^{D^{\smile}}(p, q, r)$, applying the 2D FFT inverse using $V_{ep,q}$

Here, a feature vector is formed by attaching the FDCT coefficients through a wrapping technique at every scale-based orientation, similar to how p, q is employed. To calculate the list of scales based on the size of the image $m_r \times, m_{c,}$ as follows:

$$\hat{s} = \left\lceil log_2\left((m_r, m_c)\right) - 3 \right\rceil \tag{16}$$

Each image has a size of $128 \times 128$, the $\hat{s}$ denoted as 4. It implies that every image undergoes decomposition into four levels in the Curvelet transform. Scales 2 and 3 encompass distinct sub-band information except for the initial and final scales. At angles $\alpha, \alpha + \pi$, as Curvelet generates identical coefficients. Therefore, we have eliminated a half-coefficient based on the sub-band at every scale. Therefore, the values of the feature vector's coefficients are notably elevated, which is crucial to further reduce them to isolate the prominent features.

**5.9.3. Dimensionality reduction of the feature vector.** Reducing the dimensionality of features plays a vital role in predictive modelling and tasks related to machine learning. Storing and processing feature vectors with a higher dimension requires increased storage capacity and more significant classification computational resources. Here, a commonly utilized

method named PCA has been employed in the initial stage. In the subsequent phase, linear discriminant analysis (LDA), a supervised technique capable of identifying pertinent features and distinguishing them within comparable classes, is utilised to maximize benefits [74,75]. The features acquired through PCA have been leveraged by LDA for further dimensionality reduction. Firstly, The $\widehat{D}$ denoted as a dimensional reduction feature that has reduced through the application of PCA to decrease it to M dimensions. After that, LDA was used to further reduce the dimensionality to L, resulting in the final reduced dimension being L ¡ M ¡ $D$ $\widehat{D}$ [64,69,76]. Sorting the eigenvalues of the feature list in descending order is used to pinpoint the optimal features. Then, the application of a normalized cumulative sum of variance (NCSV) technique was initiated to assess the NCSV values for each feature. By using the $i^{th}$ features of NCSV values are computed as,

$$NCSV(i) - \frac{\sum_{y-1}^{i} \beta(y)}{\sum_{y-1}^{\widehat{D}} \beta(u)} \ here \ 1 \leq i \leq \widehat{D} \tag{17}$$

Hence, $\beta(y)$ is denoted as the eigenvalue of the $y^{th}$ feature, and the dimensionality of the feature vector is $\widehat{D}$. Here, the threshold is utilized to choose $L$ as a list of eigenvector values based on NCSV.

## 5.10. IMGWO-ELM

This section describes Grey Wolf Optimization (GWO) and Extreme Learning Machine (ELM), and then introduces the proposed IMGWO-ELM algorithm.

### 5.10.1. Grey wolf optimization.
Optimization algorithms are pivotal in training machine learning models, parameter tuning, operations research, and solving real-world problems where finding the optimal solution is critical. The selection of the algorithm frequently relies on the particular attributes of the given problem, such as the nature of the objective method, the presence of constraints, and the computational resources available. There is a list of optimization methods like Gradient Descent (GD), Genetic Algorithms (GA), Simulated Annealing (SA), Ant Colony Optimization (ACO), Particle Swarm Optimization (PSO), Linear Programming, quasi-newton methods, and Gayesian Optimization (GO). GWO is a swarm intelligence method like ACO proposed by Mirjalili [77]. Its outcome is better and the best optimization technique than the genetic algorithm. It primarily mimics the leadership hierarchy of wolves. It is simulated by grouping the population of search agents into four types of individuals according to their fitness, namely α, β, Δ, and ω. Here, α is denoted as the best-fit solution, and ω is specified as a solution of least fit.

Then, α, β, and Δ guide the ω wolves to navigate the search environment for the prey, as described in Yang et al. [74]. The wolves update their position based on their encirclement of the prey, which is followed by a mathematical Equation:

$$\vec{D} = |\vec{D}. \ \vec{Y}_{P(i)} - \vec{Y}_{P(i)}| \tag{18}$$

$$\vec{Y}_{(i+1)} = \vec{Y}_{P(i)} - \vec{B}.\vec{E} \tag{19}$$

Here, $i$ denoted as the current iteration, $\vec{Y}_{p(i)}$ viewed as the current position of the prey, and $\vec{Y}_{(i)}$ shows the current position of the wolf. The vector $\vec{E}$ represents as the distance between the wolf and its prey.

Additionally, the vectors $\vec{B}$ and $\vec{D}$ are defined as:

$$\vec{B} = 2br_1 - b, \ \vec{D} = 2r_2 \tag{20}$$

$$b = 2 - i\frac{2}{maxiter} \tag{21}$$

Now, $\vec{r}_1$ and $\vec{r}_2$ have two discrete vectors with whole values in 0–1, $i$ specified as recent iteration. Finally, to achieve the best three solutions, ensure your information accurately reflects the current state of the prey; accordingly, the best three wolves have been chosen by α, β, δ [78]. Finally, the remaining wolves, having ω, modify their placement in relation to the optimal choice among the three.

The redistribution of wolves is implemented according to the subsequent equation:

$$\vec{E}_{al} = |\vec{D}.\vec{Y}_{al} - \vec{Y}|$$

(22)

$$\vec{E}_{bet} = |\vec{D}_2.\vec{Y}_{bet} - \vec{Y}|$$

(23)

$$\vec{E}_{del} = |\vec{D}_3.\vec{Y}_{del} - \vec{Y}|$$

(24)

$$\vec{Y}_1 = Y_{al} - \vec{B}_1.(\vec{E}_{al})$$

(25)

$$\vec{Y}_2 = Y_{al} - \vec{B}_2.(\vec{E}_{bet})$$

(26)

$$\vec{Y}_3 = Y_{bet} - \vec{B}_2.(\vec{E}_{del})$$

(27)

$$\vec{Y}_{i+1} = \frac{\vec{Y}_1 + \vec{Y}_2 + \vec{Y}_3}{3}$$

(28)

Here, $\vec{Y}_{al}$, $\vec{Y}_{bet}$, $\vec{Y}_{del}$, specified as the location based on α, β, δ wolves, $Y$ shows the present position of the solution $D_1, D_2, D_3, B_1, B_2, B_3$ are three randomly generated values. nomos and random allocation of hidden node parameters [79,80].

**5.10.2. Extreme learning machine.** ELM provides a simplified, streamlined, and effective approach for training neural networks, making them suitable for specific applications prioritizing quick training and simplicity. ELM has been utilized in a range of domains, including pattern recognition, image and signal processing, and regression tasks. In contrast to conventional feedforward neural networks (FFNNs) that rely on the iterative assessment of all network parameters, the ELM approach is introduced, emphasizing the use of Autofit to address regression and classification challenges. Unlike standard neural networks, Extreme Learning Machines (ELM) employ a fixed assignment of weights and biases in their hidden layers, eliminating the necessity for the iterative training process typically associated with gradient descent methods. The core idea of ELM is to randomly assign weights and biases in the hidden layer based on the input. Then, it determines the output layer weights through an analytical solution based on a linear system solver. This approach enables ELM to avoid the lengthy backpropagation process, leading to significantly quicker training durations compared to conventional neural networks. During the training phase of ELM, the network takes input data and computes the activations of the hidden layer by applying a non-linear activation function to the weighted sum of the inputs. The weights of the output layer are determined by solving using techniques like the Moore-Penrose pseudoinverse or other methods of regularization. Once training is finished, the ELM model applies the acquired weights to input features to forecast new, unseen data. Its appeal has grown due to its computational efficiency, particularly beneficial for handling extensive datasets.

Here, *Sa* specified as random distinct samples $(p_i, q_i)$, where $p_i = [p_{i1}, p_{i2}, p_{i3},..., p_{in}]^T$ and $q_i = [q_{i1}, q_{i2}, q_{i3},..., q_{im}]^T \epsilon R^m$. A conventional single-layer FFN (SLFN) with '*H*' hidden nodes specified mathematically, which follows:

$$O_I = \sum_{k=1}^{H} \beta_k \left( w_{k,} b_{k,} p_{i,} \right), \; i = 1, 2, \; 3, \; \ldots \ldots, \; S$$

(29)

In this context, $w_k$ and $b_k$ represent the parameters associated with the hidden node, encompassing its weight and bias values. The vector $b_k = [\beta_{k1}, \beta_{k2},..., \beta_{km}]^T$ serves as a representation denoting the output weight from the $k^{th}$ hidden node to the output nodes. The result linked to the $k^{th}$ node, denoted as $A(w_k, b_k, p_i)$, is equivalent to $p_i$, where $o_i$ represents the true output related to $p_i$. Additionally, the mathematical representation of $A(w_k, b_k, p_i)$ has been expressed by:

$$A\left( \omega_k, b_k, p_i \right) = a \left( \omega_k^T.p_i + b_k \right).\omega_k \in \mathcal{R}^n, b_k \in \mathcal{R}$$

(30)

Here, a(p): R '→R specified as the sigmoid function $w_k = [w_{k1}, w_{k2}, w_{k3},..., w_k]T_n$ *is* considered as the weight vector based on input, $k^{th}$ hidden nodes, and $b_k$ identified as the bias of the $k^{th}$ nodes. The SLFN can be assessed for *H* hidden nodes across *S* samples without any error. In this context, then the cost function $C = \sum_{i=1}^{S} ||(o_i - q_i)||_2 = 0$. However, there exist $(w_k, b_k)$ and $\beta_k$ like,

$$q_i = \sum_{k} = 1^H \beta_k A \left( w_k, b_k, p_i \right), \; i = 1, \; 2, \; 3, \; \cdots, S$$

(31)

In Equation 31, the specification is presented concisely as follows:

$$\hat{H}\beta = \varrho$$

(32)

Here,

$$\hat{H} = \begin{bmatrix} A(w_1, b_1 p_1) & \cdots & A(w_H, b_H p_1) \\ \vdots & \cdots & \vdots \\ A(w_i b_i p_s) & \cdots & A(w_i b_H p_s) \end{bmatrix}_{S \times H}$$

(33)

$$\beta = \begin{bmatrix} \beta_1^T \\ \beta_T^T \\ .. \\ .. \\ \beta_H^T \end{bmatrix}_{H \times m}$$

(34)

$$Q = \begin{bmatrix} q_1^T \\ q_T^T \\ .. \\ .. \\ q_H^T \end{bmatrix}_{S \times m}$$

(35)

In this context, *H* is defined as the matrix depicting the outcomes of the hidden layer in the SLFN. In Equation 32, formulated based on the linear method according to [71], characterizes the output weights associated with the model as:

$$\beta = \hat{H} \dagger Q \tag{36}$$

Here, $H^\dagger$ is specified on the Moore-Penrose (MP), specified as the opposite of $H$ [72]. Hence, the researcher is specified as MP specification inverse explained in [73] for better clarity. The algorithm explained its stages in the ELM method viewed as:

**Algorithm 1. Extreme learning machine (ELM) Algorithm pseudocode.**

```
1: Input: Specify the Training samples T^r = (p_i^r q_i^r)^N; Testing samples T^e = (P_i^r, q_i^r)^N r_i = 1;
2: Define Activation function: a(p); and Hidden Node
3: Size: H
4: Define random values for the hidden node parameters (w_k, b_k, 1≤k≤H).
5: Identify the output matrix Ĥ using Equation 33
6: Evaluate the final output weight vector β based on Equation 36
```

**5.10.3. Improved GWO-based ELM.** From Section 3.10.2, we have noticed that the ELM effectively applies its generalization characteristic to two specific parameters, namely, the weights and bias. Hence, the random initialization of hidden node parameters in ELM introduces two primary challenges, which can be readily observed [81]. Firstly, ELM exposes a significantly larger quantity of concealed neurons in contrast to conventional gradient-based algorithms. As indicated by the performance results of ELM, its speed diminishes when introduced to new test samples. Secondly, ELM focused on ill-conditioned hidden layers, resulting in vectors that evaluate the least generalization outcomes.

To address this problem, several evolutionary and swarm intelligence-based techniques have been examined and utilized [78,82,83]. These algorithms are selected because of their superior capability for conducting comprehensive searches on a global scale to address optimization challenges. In [82] presented a novel modified algorithm, like in the Evolutionary-ELM (E-ELM), an improved version of differential evolution (DE) has been utilized to optimized the hidden parameters of ELM, specifically weights & bias. The solution is obtained through inverse utilization of MP generalization. Several researchers demonstrated that E-ELM exhibits superior, quicker effectiveness than traditional algorithms. Nonetheless, a drawback of E-ELM is the necessity to fine-tune two extra parameters, precisely, mutation and crossover. Xu and Shu [83] proposed an approach that combines PSO with an ELM to optimize its parameters of ELM. Furthermore, they established a specified range within PSO to increase its overall efficacy of ELM. Han et al. [78], have an enhanced version of the ELM, referred to as IPSO-ELM, which has been shown to attain optimized SLFNs using a PSO approach.

The effectiveness of the mentioned methods is reliant on the specific parameters of the algorithm, despite ongoing efforts to optimize these parameters. Therefore, establishing appropriate values for these parameters is critical in any domain of difficulty. Additionally, inadequate tuning of parameters may lead to notable computational complexity or becoming stuck in local optima [84]. In the case of Differential Evolution (DE), two essential parameters are essential: the scaling coefficient and crossover probability. Likewise, Particle Swarm Optimization (PSO) relies on the inertia weight and acceleration factor. This research employs a modern global optimization algorithm without the need for explicit parameter settings, specifically the GWO, to address the challenges associated with improper parameter tuning. Furthermore, a novel approach, named IGWO-ELM, is introduced, merging the benefits of Grey Wolf Optimization with ELM.

**5.10.4. Improve the convergence factor.** The convergence factor emphasizes the search ability of GWO. Linear convergence is ineffective for achieving a thorough global search and tends to get trapped in local searches quickly. Usually, in global optimization algorithms, the initial phase involves exploring a large search space, which takes more time to find the optimal value. After undergoing several iterations of improvement, the algorithm typically achieves the best solution faster. Because of its similar traits to the behavior mentioned earlier, the adaptive nonlinear convergence factor linked with the exponential function intensifies this convergence phenomenon even more. The non-linear decline in value occurs as increased in list of iterations increases, exhibiting a gradual decrease in the early stages with a minor

contraction range. This enables a meticulous fine search. Subsequently, in the later stages of the iteration, the value undergoes rapid changes, facilitating an expedited search process. Using the Grey Wolf Optimizer (GWO) to dynamically find the best parameter values for the hidden nodes of the Extreme Learning Machine (ELM), and subsequently applying the Moore-Penrose pseudo-inverse for obtaining an analytical solution. Additionally, it is essential to note that the Improved GWO (IGWO) algorithm incorporates an exponential function, resulting in a modification of Equation 20. The exponential function demonstrates symmetric characteristics and features an adaptive nonlinear convergence factor as outlined in Equation 37. This allows for a variable convergence rate that can nonlinearly adjust within the range of [0, 2], based on the original linear convergence factor. To better understand how the value changes across multiple iterations, visualized in Fig 17, which graphically represents this variation.

$$a = 2 - \frac{2 * i^3}{maxiter^3} \tag{37}$$

The IMGWO+ELM system consists primarily of two components: one focused on optimizing parameters and the other on evaluating classification. The primary objective of IGWO is to enhance classification accuracy while ensuring that input weights, hidden bias remain within the specified range. This is done to enhance the convergence characteristics of the ELM. The subsequent passage provides a brief overview of the various stages involved in IMGWO-ELM.

- Initially, each candidate solution is randomly initialized within the population. Each solution consists of a set of input weights and a bias, organized as follows:

$$\omega_j = \left[\omega^i_{11}, \omega^i_{1l}, \omega^i_{2l}, \omega^i_{22}, \cdots, \omega^i_{2l}, \cdots, \omega^i_{H1}, \omega^i_{H2}, \cdots, \omega^i_{Hl}\right]; \; -1 \leq \omega \leq 1 \; b_j = \left[b_1, b_2, \cdots, b_H\right]; \; -1 \leq b \leq 1 \; C_j = [\omega_j, b_j] \tag{38}$$

- Evaluate the classification accuracy and output weights for each potential solution. Specifically, assess the fitness value, which indicates classification accuracy, using the validation set to avoid overfitting issues.

- Determine the three most effective outcomes (α, β, δ). Revise the outcomes by making updates in Equation 29.

- Generate innovative solutions by employing the fitness value in the following manner:

$$C_J(k + 1) = \begin{cases} c^i_j(k) & if \; c^i_j(k) > \int c_j(k)) \\ c_j(k) & otherwise \end{cases} \tag{39}$$

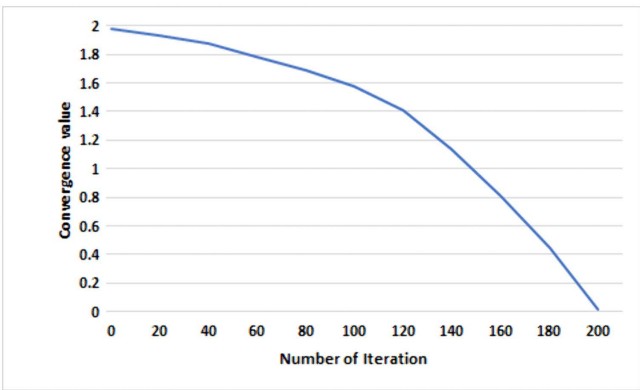

**Fig 17. The Graph of the convergence factor with number of iteration of the employed model.**

During this context, $f(c_j(k))$ and $c_j^i(k)$ denote the fitness of the $j^{th}$ candidate solution and its corresponding updated solution based on the $k^{th}$ iteration, respectively.

- Search for occurrences of cases beyond bounds in each of the recently created solutions and limit them to a range of [-1, 1] using the following method:

$$c_j = \begin{cases} -1 & if c_j (k+1) < -1 \\ 1 & if\ c_j (k+1) > 1 \end{cases} \tag{40}$$

- Continue iterating through steps (III) to (V) until you reach the termination state. The termination state corresponds to the target fitness value that must be achieved within a specified number of iterations. Once this target is met, the optimized parameters of the hidden nodes are crucial and will be applied to the testing sample to evaluate the overall performance of the implemented scheme.

The algorithm 2 outlining the deployed scheme is presented in the pseudo-code.

**Algorithm 2. The Proposed model (FDCT-WRP + PCA + LDA+IMGWO) Algorithm pseudocode.**

```
1: For every extracted ROI Implement FDCT-WRP methods. Concatenate all features vectors and create
   the feature vector of dimension F.
2: End For
3: Deploy PCA+LDA technique to reduce feature vector from F→k and then k→f.
4: Use five-fold stratified cross-validation on the trimmed feature set to generate the training and
   testing datasets.
5: Train ELM model by utilizing improved gray wolf optimization algorithm.
6: Determine the most favorable values for the parameters associated with hidden nodes, specifically
   the weights and biases.
7: Establish the output weights of ELM using the refined input weights and biases that have been
   computed.
8: Evaluate the performance results of IMGWO-ELM on the test set.
```

## 6. Conclusion and future work

Our proposed manuscript presents an improved explainability framework called GlaucoXAI (Glaucoma Explainable Artificial Intelligence). GlaucoXAI aims to enhance our understanding of deep learning networks' behavior by utilizing advanced visualization techniques, such as attention maps. As a post hoc tool, GlaucoXAI can be applied to any existing deep neural models, offering significant insights into their operations.

Our two case studies highlight the importance of integrating explainable AI (XAI) techniques in medical image analysis. Additionally, GlaucoXAI facilitates the Extreme Learning Machine (ELM) classifier for glaucoma detection. Our findings underscore the crucial role of XAI in medical imaging tasks, helping to improve the comprehensibility of machine learning models and accelerating their adoption by medical professionals.

This paper introduces a sophisticated computer-aided design (CAD) model specifically designed for categorizing glaucoma and healthy images. The model effectively identifies relevant features in fundus images by utilizing a fast discrete curvelet transform with a wrapping (FDCT-WRP) process. To enhance feature reduction, we apply a combination of Principal Component Analysis (PCA) and Linear Discriminant Analysis (LDA), resulting in a set of reduced and more prominent features.

Subsequently, the CAD model employs the IMGWO-ELM, a faster learning algorithm, to train the Single-Layer Feed-forward Network (SLFN). We rigorously evaluate the CAD model's classification performance across two standard fundus image datasets. The experimental results demonstrate that the proposed CAD model achieves superior classification performance with fewer features compared to existing models.

In future research, we plan to test the efficacy of our approach for generalization across various imaging modalities. Another potential avenue of exploration is the hybridization of ELM with a less parameter-based optimization algorithm, assessing its effectiveness in multi-class classification tasks. Additionally, the paper suggests considering deep learning algorithms as viable alternatives to the proposed model.

Our future research will also focus on quantitatively evaluating XAI methods. This evaluation aims to assess the effectiveness of sensitivity maps generated by these methods and their correlation with deep learning accuracy metrics. We plan to conduct further experiments in the realm of multi-modal glaucoma detection to enhance our understanding. Moreover, we aim to investigate the potential for extracting quantitative features, such as tumor volume and centroid, from these explanation methods.

## Acknowledgments

We sincerely thank the Department of Computer Science at C.V. Raman University and Nagaland University for providing laboratory facilities.

## Author contributions

**Conceptualization:** Debendra Muduli, Santosh Kumar Sharma, Sujata Dash, Bernardo Lemos, Saurav Mallik.

**Data curation:** Debendra Muduli, Santosh Kumar Sharma, Sujata Dash, Bernardo Lemos, Saurav Mallik.

**Formal analysis:** Debendra Muduli, Sujata Dash, Bernardo Lemos.

**Investigation:** Santosh Kumar Sharma, Sujata Dash, Bernardo Lemos.

**Methodology:** Debendra Muduli, Santosh Kumar Sharma.

**Project administration:** Bernardo Lemos.

**Resources:** Debendra Muduli, Saurav Mallik.

**Software:** Santosh Kumar Sharma, Sujata Dash, Saurav Mallik.

**Supervision:** Sujata Dash, Bernardo Lemos, Saurav Mallik.

**Validation:** Sujata Dash, Bernardo Lemos, Saurav Mallik.

**Visualization:** Debendra Muduli, Santosh Kumar Sharma, Saurav Mallik.

**Writing – original draft:** Debendra Muduli, Santosh Kumar Sharma, Sujata Dash.

**Writing – review & editing:** Sujata Dash, Bernardo Lemos, Saurav Mallik.

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
