## [Decision Letter · Decision Letter 0]

7 Sep 2025

PCOMPBIOL-D-25-00962

Explainable AI-Driven Diagnosis Model for Early Glaucoma Detection Using Grey-Wolf Optimized Extreme Learning Machine Approach

PLOS Computational Biology

Dear Dr. Mallik,

Thank you for submitting your manuscript to PLOS Computational Biology. After careful consideration, we feel that it has merit but does not fully meet PLOS Computational Biology's publication criteria as it currently stands. Therefore, we invite you to submit a revised version of the manuscript that addresses the points raised during the review process.

Please submit your revised manuscript within 60 days Nov 07 2025 11:59PM. If you will need more time than this to complete your revisions, please reply to this message or contact the journal office at ploscompbiol@plos.org. Please include the following items when submitting your revised manuscript:

We look forward to receiving your revised manuscript.

Kind regards,

Varun Dutt, Ph.D

Academic Editor

PLOS Computational Biology

Jennifer Flegg

Section Editor

PLOS Computational Biology

**Additional Editor Comments:**

Reviewer #1:

Reviewer #2:

**Journal Requirements:**

Potential Copyright Issues:

i) Figure 1. Please confirm whether you drew the images / clip-art within the figure panels by hand. If you did not draw the images, please provide (a) a link to the source of the images or icons and their license / terms of use; or (b) written permission from the copyright holder to publish the images or icons under our CC BY 4.0 license. Alternatively, you may replace the images with open source alternatives. See these open source resources you may use to replace images / clip-art:

6) Kindly revise your competing statement to align with the journal's style guidelines: 'The authors declare that there are no competing interests.'

**Reviewers' comments:**

Reviewer's Responses to Questions

**Comments to the Authors:**

Reviewer #1: Thank you for your submission. This manuscript introduces a hybrid AI-based Computer-Aided Diagnosis (CAD) system for early glaucoma detection using fundus images, integrating an Improved Grey Wolf Optimized Extreme Learning Machine (IMGWO-ELM) with a custom explainability framework termed GlaucoXAI. The topic is timely, well-motivated, and addresses a clinically significant challenge—balancing performance and interpretability in AI-based medical diagnostics.

Below is a detailed review of the strengths, limitations, and recommendations to improve the manuscript:

Strengths:

1. Novelty and Relevance

The hybrid architecture combining FDCT-WRP, PCA+LDA, and IMGWO-ELM is a thoughtful and innovative design.

Integration of seven Explainable AI methods into the GlaucoXAI framework enhances clinical trust, a major concern in real-world adoption of AI systems.

2. Strong Experimental Validation

Results on two publicly available datasets (G1020 and ORIGA) show strong classification performance (93.87% and 95.38% accuracy respectively).

Use of 10×5-fold stratified cross-validation adds robustness to the evaluation.

3. Interpretability Emphasis

The manuscript goes beyond mere performance by incorporating sensitivity maps and visual explanations using VG, IG, Grad-CAM, and other XAI techniques.

4. Resource-Efficient Model

The use of ELM allows for fast learning with low computational cost, which is valuable for deployment in resource-constrained settings.

Limitations and Areas for Improvement:

1. Language and Grammar:

The manuscript contains numerous grammatical and syntactic errors, awkward phrasing, and redundant wording that significantly affect readability.

A thorough professional language editing is strongly recommended.

2. Explainability Evaluation Is Superficial

While visual outputs are shown, no quantitative evaluation of the explanation quality is provided (e.g., overlap with expert annotations, localization metrics, or user studies).

A clinician-centric validation of interpretability would strengthen the paper considerably.

3. Limited Dataset and Generalizability

The study relies only on G1020 and ORIGA, both from similar sources. The lack of external validation limits generalization.

Future work should aim to include multi-center or device-diverse data to demonstrate robustness.

4. No Code Availability

The manuscript does not provide code or implementation details beyond pseudocode. This limits reproducibility, which is a key requirement of PLOS.

Authors should deposit source code and trained model weights in a public repository (e.g., GitHub).

5. Pipeline Complexity

The combination of FDCT-WRP, PCA+LDA, IMGWO, and ELM introduces multiple layers of abstraction and potential tuning challenges.

An ablation study to quantify the individual contribution of each component would improve scientific clarity.

6. Figures and Notation Issues

Several figures (e.g., Figure 3, convergence graph) are not clearly referenced or explained in the main text.

Mathematical expressions are inconsistently formatted and sometimes contain LaTeX remnants (e.g., \rightarrow, \alpha).

Suggestions for Improvement:

1. Perform a comprehensive language and formatting review.

2. Include quantitative evaluation of XAI methods, possibly via comparison with clinician annotations.

3. Share the full codebase and model artifacts in a public repository.

4. Discuss computational cost in more detail—time per image, resources used, etc.

5. Consider an ablation study or include a simpler baseline like CNN+Grad-CAM to compare against.

6. If possible, validate on an external dataset or simulate domain shift scenarios.

Final Recommendation:

Accept with Major Revisions

The paper has strong scientific merit and introduces a meaningful contribution. However, before it can be considered for publication, the above-mentioned concerns—particularly around language, reproducibility, and explainability validation—must be addressed.

Reviewer #2: 1 The proposed approach FDCT-WRP+PCA+LDA+IMGWO+ELM (Proposed Model) must add the time complexity as compared to simpler approaches I think , what is the time complexity of this approach, is it justifiable to add the complexity for the sake of increasing the performance

2. PCA and LDA are very old and outdated approaches , please justify why they have been used

3. what is the improvement in performance when so much modifications are suggested , what % changes in the results have been observed

4. We are unable to identify the exact changes made in GWO algorithm ? please discuss in detail

5. Please list the exact features in tabular format during each phase , which of the features are selected and which are rejected , any why , if possible

6. Addressing issues related to scalability, interoperability, and regulatory compliance would enhance the practical relevance of the conclusions.

7. Although the paper mentions the need for further validation on larger datasets, it could benefit from a more comprehensive discussion on specific avenues for future research and development. Identifying specific research gaps or unresolved questions would provide a clearer roadmap for advancing the field of this disease, its diagnosis and monitoring using deep/machine learning techniques.

8. These few recent state-of-the-art studies should be discussed where similar features , ML and feature selection have been employed for GLAUCOMA diagnosis ( A three-stage novel framework for efficient and automatic glaucoma classification from retinal fundus images; An artificial intelligence-based smart system for early glaucoma recognition using OCT images ; Histogram of oriented gradients (HOG)-based artificial neural network (ANN) classifier for glaucoma detection)

9. How one can ensure that the proposed system is generalizable and will repeat the same performance on other datasets also.

**Have the authors made all data and (if applicable) computational code underlying the findings in their manuscript fully available?**

Reviewer #1: **No:** Partially.

The authors state in the manuscript's Data Availability Statement that the datasets used (G1020 and ORIGA) are publicly available and properly referenced. However, no source code for the proposed methodology (including FDCT-WRP feature extraction, PCA+LDA processing, IMGWO optimization, or the GlaucoXAI explainability module) has been shared in the manuscript or as supplementary material.

To fully comply with the PLOS Data Policy, the authors should make their computational code and trained model weights available via a public repository.

Reviewer #2: Yes

PLOS authors have the option to publish the peer review history of their article (what does this mean?). If published, this will include your full peer review and any attached files.

Reviewer #1: **Yes:** Shruti Kaushik

Reviewer #2: No

**Figure resubmission:**
---

## [Decision Letter · Decision Letter 1]

11 Feb 2026

PCOMPBIOL-D-25-00962R1

Explainable AI-Driven Diagnosis Model for Early Glaucoma Detection Using Grey-Wolf Optimized Extreme Learning Machine Approach

PLOS Computational Biology

Dear Dr. Mallik,

Thank you for submitting your manuscript to PLOS Computational Biology. After careful consideration, we feel that it has merit but does not fully meet PLOS Computational Biology's publication criteria as it currently stands. Therefore, we invite you to submit a revised version of the manuscript that addresses the points raised during the review process.

We look forward to receiving your revised manuscript.

Kind regards,

Varun Dutt, Ph.D

Academic Editor

PLOS Computational Biology

Jennifer Flegg

Section Editor

PLOS Computational Biology

**Journal Requirements:**

1) Your manuscript is missing the following sections: Results, and Methods.  Please ensure all required sections are present and in the correct order. Make sure section heading levels are clearly indicated in the manuscript text, and limit sub-sections to 3 heading levels. An outline of the required sections can be consulted in our submission guidelines here:

2) In the online submission form, you indicated that your data will be submitted to a repository upon acceptance. We strongly recommend all authors deposit their data before acceptance, as the process can be lengthy and hold up publication timelines. Please note that, though access restrictions are acceptable now, your entire minimal dataset will need to be made freely accessible if your manuscript is accepted for publication. This policy applies to all data except where public deposition would breach compliance with the protocol approved by your research ethics board. If you are unable to adhere to our open data policy, please kindly revise your statement to explain your reasoning and we will seek the editor's input on an exemption.

**Reviewers' comments:**

Reviewer's Responses to Questions

**Comments to the Authors:**

Reviewer #1: Thank you for submitting the revised version of your manuscript, “Explainable AI-Driven Diagnosis Model for Early Glaucoma Detection Using Grey-Wolf Optimized Extreme Learning Machine Approach.” The revision reflects effort and meaningful engagement with the reviewers’ comments. The manuscript has improved significantly in terms of technical rigor, reproducibility, methodological clarity, and compliance with PLOS requirements.

Before the manuscript can be recommended for publication, I encourage the authors to address the following minor but important points, primarily related to clarity, consistency, and the strength of the explainability validation.

1. Explainability Evaluation: Clarify Scope and Evidence

The manuscript now goes beyond purely qualitative visualizations and claims to include quantitative and clinician-centric evaluation of explainability, which is a positive step. However, the current presentation lacks sufficient methodological detail to fully assess the robustness of these claims.

Please clarify the following explicitly in the manuscript:

How many clinicians participated in the evaluation?

What was the annotation protocol (e.g., optic disc/cup, RNFL regions, disease-relevant ROIs)?

Which quantitative metrics were used to compare saliency maps with expert annotations (e.g., Dice coefficient, IoU, localization error)?

Were multiple annotators involved, and if so, was inter-rater variability assessed?

If the clinician study or quantitative evaluation is limited in scale, this should be clearly stated as a limitation, rather than implied as a comprehensive validation. Strengthening transparency here will significantly improve the credibility of the GlaucoXAI framework.

2. Domain Consistency and Residual Artifacts

There are several places in the manuscript, particularly within the explainability discussion, where references to glioma, brain imaging, MRI, or FLAIR images appear. These seem to be residual artifacts from prior work and are not appropriate in the context of glaucoma fundus image analysis.

Action required:

Carefully review the manuscript to remove or replace all references that are unrelated to retinal fundus imaging and glaucoma.

Ensure that all XAI discussions, examples, and interpretations are strictly grounded in ophthalmic imaging and glaucoma-specific pathology.

This correction is important for conceptual coherence and editorial quality.

3. Language, Redundancy, and Editorial Polish

The overall readability has improved compared to the original submission, but the manuscript still contains:

Redundant explanations (particularly in the methodology and results sections),

Overly verbose figure captions,

Occasional awkward phrasing and grammatical inconsistencies.

Recommendations:

Perform a final language and style pass to improve conciseness and flow.

Reduce repetition across sections describing FDCT-WRP, PCA+LDA, and IMGWO-ELM.

Streamline figure captions so they explain what is shown without repeating large portions of the main text.

This will help align the manuscript with the editorial standards expected by PLOS Computational Biology.

4. Positioning of Classical Methods (PCA, LDA, ELM)

Your justification for using PCA, LDA, and ELM, despite their classical nature, is reasonable and aligns with your goals of efficiency, interpretability, and deployment feasibility. However, this rationale would benefit from a more explicit positioning statement early in the manuscript.

Suggested improvement:

Clearly emphasize that the novelty lies not in individual components, but in their synergistic integration with IMGWO and a multi-method XAI framework, rather than in proposing new standalone algorithms.

This will help pre-empt potential criticism regarding the use of established techniques.

5. Generalizability and External Validation

The added discussion on dataset limitations, domain-shift simulations, and future multi-center validation is appropriate and sufficient for this revision. No additional experiments are required at this stage. However, ensure that claims about robustness and generalization are carefully worded and do not overstate the current empirical evidence.

Addressing the minor revisions outlined above, particularly clarifying the explainability evaluation, removing domain inconsistencies, and improving editorial polish, will make the manuscript suitable for publication.

Reviewer #2: Accepted from my side

**Have the authors made all data and (if applicable) computational code underlying the findings in their manuscript fully available?**

Reviewer #1: Yes

Reviewer #2: Yes

PLOS authors have the option to publish the peer review history of their article (what does this mean?). If published, this will include your full peer review and any attached files.

Reviewer #1: **Yes:** Dr. Shruti Kaushik

Reviewer #2: No

**Figure resubmission:**
---

## [Decision Letter · Decision Letter 2]

19 Apr 2026

Dear Dr. Mallik,

We are pleased to inform you that your manuscript 'Explainable AI-Driven Diagnosis Model for Early Glaucoma Detection Using Grey-Wolf Optimized Extreme Learning Machine Approach' has been provisionally accepted for publication in PLOS Computational Biology.

Best regards,

Varun Dutt, Ph.D

Academic Editor

PLOS Computational Biology

Jennifer Flegg

Section Editor

PLOS Computational Biology

Based upon the revisions, the manuscript may be accepted in its current form.

Reviewer's Responses to Questions

**Comments to the Authors:**

Reviewer #1: Thank you for your careful and thoughtful revision of the manuscript. The paper has improved significantly in terms of clarity, structure, and methodological transparency. I appreciate the effort taken to address the previous comments, particularly the improvements in domain consistency, the clearer positioning of the methodological contributions, and the addition of a limitation discussion.

The clarification of the explainability evaluation, including the involvement of clinicians and the annotation framework, is a positive step. However, as currently presented, the evaluation remains primarily qualitative. While this is acceptable for the current study, it is important that the manuscript consistently frames the explainability validation as preliminary rather than comprehensive. Please ensure that this positioning is clearly and consistently reflected throughout the text, especially in the abstract, results, and discussion sections.

Additionally, a final light editorial pass for language conciseness and flow would further strengthen the manuscript.

Overall, the manuscript is now well-prepared and suitable for publication.

**Have the authors made all data and (if applicable) computational code underlying the findings in their manuscript fully available?**

Reviewer #1: Yes

PLOS authors have the option to publish the peer review history of their article (what does this mean?). If published, this will include your full peer review and any attached files.

Reviewer #1: **Yes:** Dr Shruti Kaushik

---

## [Editor Report · Acceptance letter]

PCOMPBIOL-D-25-00962R2

Explainable AI-Driven Diagnosis Model for Early Glaucoma Detection Using Grey-Wolf Optimized Extreme Learning Machine Approach

Dear Dr Mallik,

I am pleased to inform you that your manuscript has been formally accepted for publication in PLOS Computational Biology. Your manuscript is now with our production department and you will be notified of the publication date in due course.

With kind regards,

Anita Estes
